# HeroLT: Benchmarking Heterogeneous Long-Tailed Learning

## Abstract

Long-tailed data distributions are prevalent in a variety of domains, including e-commerce, finance, biomedical science, and cyber security. In such scenarios, the performance of machine learning models is often dominated by the head categories, while the learning of tail categories is significantly inadequate. Given abundant studies conducted to alleviate the issue, this work aims to provide a systematic view of long-tailed learning with regard to three pivotal angles: (A1) the characterization of data long-tailedness, (A2) the data complexity of various domains, and (A3) the heterogeneity of emerging tasks. To achieve this, we develop the most comprehensive (to the best of our knowledge) long-tailed learning benchmark named HeroLT, which integrates 15 state-of-the-art algorithms and 6 evaluation metrics on 16 real-world benchmark datasets across 5 tasks from 3 domains. HeroLT with novel angles and extensive experiments (304 in total) enables researchers and practitioners to effectively and fairly evaluate newly proposed methods compared with existing baselines on varying types of datasets. Finally, we conclude by highlighting the significant applications of long-tailed learning and identifying several promising future directions. For accessibility and reproducibility, we open-source our benchmark HeroLT and corresponding results at `https://anonymous.4open.science/r/HeroLT-9746/`.

## 1 Introduction

In the era of big data, many high-impact domains, such as e-commerce (Zhang et al., 2021e; Wang et al., 2016), finance (Wang et al., 2019), biomedical science (Zhang et al., 2021a; Ju et al., 2021), and cyber security (Kuypers et al., 2016; Wang et al., 2020a), naturally exhibit complex and long-tailed data distributions, where a few head categories[1] are well-studied with abundant data, while the massive tail categories are under-explored with scarce data. To name a few, in financial transaction networks, the majority of transactions fall into a few head classes that are considered normal, like wire transfers and credit card payments. However, there are a large number of tail classes corresponding to a variety of fraudulent transaction types, such as synthetic identity transactions and money laundering. Although fraudulent transactions rarely occur, detecting them is essential for preventing unexpected financial loss (Singleton & Singleton, 2010; Akoglu et al., 2015). Another example is antibiotic resistance genes, which can be classified based on the antibiotic class they confer to and their transferable ability. Genes with mobility and strong human pathogenicity may not be detected frequently and can be viewed as tail classes, but these resistance genes have the potential to be transmitted from the environment to bacteria in humans, thereby posing an increasing global threat to human health and the treatment of critical diseases (Arango-Argoty et al., 2018; Li et al., 2021b).

To address long-tailed problems, massive long-tailed learning studies have been conducted in recent years. Lin et al. (2017) propose focal loss, a cost-sensitive method, to effectively address the data imbalance by reshaping the standard cross-entropy loss. Zhao et al. (2021) solves the problem by interpolating tail node embeddings and generating new edges to augment the original graph. The advancements in tackling long-tailed problems drove the publication of several studies on the survey of long-tailed problems (Fang et al., 2023; Fu et al., 2022; Zhang et al., 2021f; 2023; Yang et al., 2022), which generally categorize existing works by different types of learning algorithms (*e.g.*, re-sampling (Chawla et al., 2002; Liu et al., 2009), cost-sensitive (Cao et al., 2019; Wang et al.,

---

[1]Long-tailed problems occur in labels or input data (like degrees of nodes), collectively named as category.

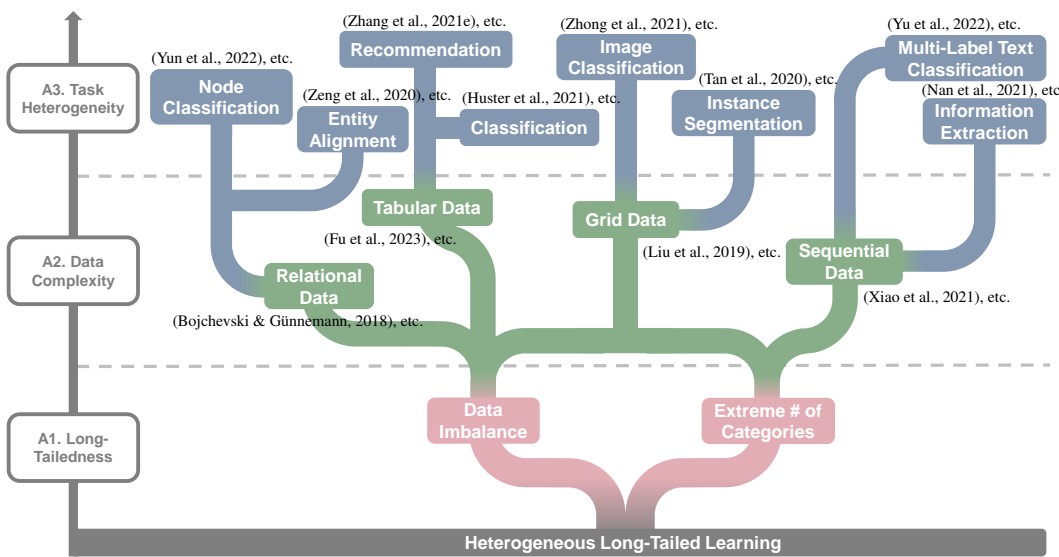

Figure 1: The systematic view of heterogeneous long-tailed learning concerning three pivotal angles, including long-tailedness (colored in red), data complexity (green), and task heterogeneity (blue).

2021a), transfer learning (Liu et al., 2021a; Li et al., 2021a), decoupled training (Zhang et al., 2021c; Kang et al., 2020), and meta learning (Jamal et al., 2020; Wang et al., 2017; Shu et al., 2019)).

Despite tremendous progress, some pivotal questions largely remain unresolved, *e.g.*, *how can we characterize the extent of long-tailedness of given data? how do long-tailed algorithms perform with regard to different tasks on different domains?* To fill the gap, this work aims to provide a systematic view of long-tailed learning with regard to three pivotal angles in Fig. 1: **(A1) the characterization of data long-tailedness:** long-tailed data exhibits a highly skewed data distribution and an extensive number of categories; **(A2) the data complexity of various domains:** a wide range of complex domains may naturally encounter long-tailed distribution, *e.g.*, tabular data, sequential data, grid data, and relational data; and **(A3) the heterogeneity of emerging tasks:** it highlights the need to consider the applicability and limitations of existing methods on heterogeneous tasks. With three major angles in long-tailed learning, we design extensive experiments that conduct 15 state-of-the-art algorithms and 6 evaluation metrics on 16 real-world benchmark datasets across 5 tasks from 3 domains.

**Key Takeaways:** Through extensive experiments (see Sec. 3), we find (1) most works mainly focus on data imbalance while paying less attention to the extreme number of categories in the long-tailed distribution; (2) surprisingly none of the algorithms statistically outperforms others across all tasks and domains, emphasizing the importance of algorithm selection in terms of scenarios.

In general, we summarize the main contributions of HEROLT as below:
- **Comprehensive Benchmark.** We conduct a comprehensive review and examine long-tailed learning concerning three pivotal angles: (A1) the characterization of data long-tailedness, (A2) the data complexity of various domains, and (A3) the heterogeneity of emerging tasks.
- **Novel Metric.** We investigate previous quantitative metrics of long-tailedness (LT) and introduce Pareto-LT Ratio as a new metric for better measuring datasets.
- **Insights and Future Directions.** With comprehensive results, our study highlights the importance of characterizing the extent of long-tailedness and algorithm selection while identifying open problems and opportunities to facilitate future research.
- **The Open-Sourced Toolbox.** We provide a fair and accessible performance evaluation of 15 state-of-the-art methods on multiple benchmark datasets using accuracy-based and ranking-based evaluation metrics at `https://anonymous.4open.science/r/HeroLT-9746/`.

## 2 HEROLT: BENCHMARKING HETEROGENEOUS LONG-TAILED LEARNING

### 2.1 PRELIMINARIES AND PROBLEM DEFINITION

Here we provide a general definition of long-tailed learning as follows. Given a long-tailed dataset $\mathcal{D} = \{\mathbf{x}_1, \ldots, \mathbf{x}_n\}$ of $n$ samples from $C$ categories, let $\mathcal{Y}$ denote the label set of category, where a

Figure 2: Illustrative figures of a synthetic long-tail distributed data. (a) long-tailed distribution of categories. (b) Visualization of obeying of Assumption 1, 2. (c) Visualization of violating of Assumption 1. (d) Visualization of violating of Assumption 2. (e) Visualization of violating of Assumption 1, 2.

sample $\mathbf{x}_i$ may belong to one or more categories. For example, in image classification, one sample belongs to one category; in document classification, one document is often associated with multiple categories (*i.e.*, topics). For simplicity, we represent $\mathcal{D} = \{\mathcal{D}_1, \ldots, \mathcal{D}_C\}$, where $\mathcal{D}_c$, $c = 1, \ldots, C$ denotes the subset of samples belong to category $c$, and the size of each category $n_c = |\mathcal{D}_c|$ following a descending order. As an instantiation, a synthetic long-tailed dataset is given in Fig. 2(a). Categories with massive samples refer to head categories, while categories with only a few samples refer to tail categories. Without the loss of generality, we make the following assumptions in long-tailed learning regarding the decision region of head and tail categories on the embedding space.

**Assumption 1** (Smoothness Assumption for Head Category). *Given a long-tailed dataset, the distribution of the decision region of each head category is sufficiently smooth.*

**Assumption 2** (Compactness Assumption for Tail Category). *Given a long-tailed dataset, the tail category samples can be represented as a compact cluster in the feature space.*

These assumptions ensure that tail categories are identifiable and meaningful. If Assumption 1 of smoothness is violated, then it is challenging to identify the head category clearly. For example in Fig. 2(c), it is difficult to depict the decision region of the head category if the smoothness is violated. Similarly, if Assumption 2 of compactness is violated, as seen in Fig. 2(d), it is difficult to determine whether it is noise data or data from tail category. Based on the above assumptions, we give the definition of long-tailed learning.

**Problem 1. *Long-Tailed Learning.***

***Given:*** *a training set $\mathcal{D}$ of $n$ samples from $C$ distinct categories following long-tailed distribution, and the label set of category $\mathcal{Y}$. The data follows long-tailed distribution, i.e., the frequency of the categories is approximated as $\lim_{y \to \infty} e^{ty} P(Y > y) = \infty, \forall t > 0$, where $Y$ is a random variable.*
***Find:*** *a function $f : \mathcal{X} \to \mathcal{Y}$ that gives accurate label predictions on both head and tail categories.*

## 2.2 BENCHMARK ANGLES IN HEROLT

**Angle 1: Long-Tailedness in Terms of Data Imbalance and Extreme Number of Categories.** While extensive public datasets and numerous algorithms are available for benchmarking, these datasets or algorithms often exhibit varying extents of long-tailedness or focus on solving specific characteristics of long-tailed problems, making it challenging to select appropriate datasets and baselines for evaluating new long-tailed algorithms. Two key properties of long-tail distributed data lie in highly skewed data distribution and an extreme number of categories. The former introduces a significant difference in sample sizes between head and tail categories, resulting in a bias towards the head categories(Zhang et al., 2023), while the latter poses challenges in learning classification boundaries due to the increasing number of categories (Mittal et al., 2021; Zhang & Zhou, 2013).

To measure the long-tailedness of a dataset, we introduce three metrics in Table 1. Firstly, a commonly used metric is imbalance factor (IF) (Cui et al., 2019), and a value closer to 1 means a more balanced dataset. Secondly, Yang et al. (2022) propose using Gini coefficient as a metric to quantify long-tailedness. It ranges from 0 to 1, where a smaller value indicates a more balanced dataset. IF quantifies data imbalance between the most majority and the most minority categories, while Gini coefficient measures overall imbalance, unaffected by extreme samples or absolute data size. However, the two metrics pay more attention to data imbalance and may not reflect the number of categories. Therefore, we propose using Pareto-LT Ratio to jointly evaluate the two aspects of the data long-tailedness. Intuitively, the higher the skewness of the data distribution, the higher Pareto-LT Ratio may be; the more number of categories, the higher the value of Pareto-LT Ratio.

In light of three long-tailedness metrics, we gain a better understanding of which datasets and baselines to consider when evaluating a newly proposed algorithm.

Table 1: Three metrics for measuring long-tailedness of datasets. $Q(p) = min\{y : Pr(\mathcal{Y} \leq y) = p, 1 \leq y \leq C\}$ is the quantile function of order $p \in (0, 1)$ for $\mathcal{Y}$.

| Metric name | Computation | Description |
|---|---|---|
| Imbalance Factor (Cui et al., 2019) | $n_1/n_C$ | Ratio to the size of largest and smallest categories. |
| Gini Coefficient (Yang et al., 2022) | $\frac{\sum_{i=1}^{C}\sum_{j=1}^{C}|n_i-n_j|}{2nC}$ | Relative mean absolute differences between category sizes. |
| Pareto-LT Ratio | $\frac{C-Q(0.8)}{Q(0.8)}$ | The number of categories of the last 20% samples to the number of categories of the rest 80% samples. |

- When all three metrics (Imbalance factor, Gini coefficient, Pareto-LT Ratio) have large values, it indicates a dataset with a severe long-tailed distribution, necessitating an algorithm that addresses both data imbalance and the extreme number of categories.
- If Gini coefficient and IF are relatively small, but Pareto-LT Ratio is large, the main challenges of the dataset may lie in the extreme number of categories, as exemplified by the experiments in Sec. 3.4. In such cases, methods as extreme classification (Saini et al., 2021; Bhatia et al., 2015) and meta learning (Jamal et al., 2020; Wang et al., 2017; Shu et al., 2019) would be preferred.
- When Gini coefficient and IF are large, but Pareto-LT Ratio is relatively small, it suggests that the challenges of data imbalance may be more significant than the number of categories. Algorithms employing techniques like re-sampling (Chawla et al., 2002; Liu et al., 2009) or re-weighting (Cao et al., 2019; Wang et al., 2021a) may already effectively handle data imbalance, as exemplified by the experiments in Sec. 3.5.
- If all three metrics are small, the extent of the long-tailedness of the dataset may be small, and ordinary machine learning methods may achieve decent performance.

We give a detailed evaluation of long-tailedness metrics (Appx.B.1), and show the long-tailedness of benchmark datasets (Table 2). We analyze whether long-tailed algorithms are specifically designed to address data imbalance and an extreme number of categories (Appx.B.2), and provide a comprehensive experimental analysis (Sec.3).

**Angle 2: Data Complexity with 16 Datasets across 3 Domains.** Most existing long-tailed benchmarks mainly focus on long-tailed image datasets (Fang et al., 2023; Fu et al., 2022; Zhang et al., 2021f; 2023; Yang et al., 2022). However, in real-world applications, various types of data including tabular data, grid data, sequential data, and relational data, may face long-tailed problems. To fill this gap, we consider data complexity based on the types of data with long-tailed distribution.

- **Tabular data** comprises samples (rows) with the same set of features (columns) and is used in practical applications (*e.g.*, medicine, finance, and e-commerce). Specifically, Glass (German, 1987) is a tabular dataset with feature attributes, where the number of samples in different categories follows a long-tailed distribution. In addition to node connections, graph data such as Amazon (McAuley et al., 2015) often include node attributes, which can be regarded as tabular data.
- **Sequential data** denotes that data points in the dataset are dependent on the other points. A common example is a time series such as S&P 500 Daily Changes (Huster et al., 2021), where the input (price over date) shows a long-tailed distribution. Another example of sequential data is text composed of manuscripts, messages, and reviews, such as Wikipedia (You et al., 2019) containing Wikipedia articles with the long-tailed distribution of Wikipedia categories.
- **Grid data** records regularly spaced samples over an area. Images can be viewed as grid data by mapping grid cells onto pixels one for one. The labels of images often exhibit long-tailed distribution, as observed in commonly used datasets such as ImageNet (Liu et al., 2019), iNatural (Van Horn et al., 2018), and LVIS (Gupta et al., 2019). Furthermore, HOI categories in HICO-DET (Chao et al., 2015) and V_COCO (Gupta & Malik, 2015), which are designed for human-object interaction detection in images, follow the long-tailed distribution. Videos can be regarded as a combination of sequential and grid data. In INSVIDEO dataset(Li et al., 2019) for micro-video, hashtags exhibit long-tailed distribution. While the NYU-Depth dataset (Silberman et al., 2012), which is composed of video sequences as recorded by depth cameras, exhibits a long-tailed per-pixel depth.
- **Relational data** organizes data points with defined relationships. One specific type of relational data is represented by graphs, which consist of nodes connected by edges. Therefore, in addition to long-tailed distributions in node classes, node degrees, referring to the number of edges connected to a node, may exhibit a long-tailed distribution (Liu et al., 2021b). It is frequently observed that the majority of nodes have only a few connected edges, while only a small number of nodes have a large number of connected edges, as seen in datasets like Cora (Bojchevski & Günnemann, 2018),

Table 2: Datasets available in HEROLT benchmark. The values in this table correspond to input data.

| Dataset | Data | Data Statistics | | | Long-Tailedness | | |
| --- | --- | --- | --- | --- | --- | --- | --- |
| | | # of Categories | Size | # of Edges | IF | Gini | Pareto |
| Glass (German, 1987) | Tabular | 6 | 214 | - | 8 | 0.380 | 0.500 |
| Abalone (Nash & Ford, 1995) | Tabular | 28 | 4177 | - | 689 | 0.172 | 0.333 |
| EURLEX-4K (You et al., 2019) | Sequential | 3,956 | 15,499 | - | 1,024 | 0.342 | 3.968 |
| AMAZONCat-13K (You et al., 2019) | Sequential | 13,330 | 1,186,239 | - | 355,211 | 0.327 | 20.000 |
| Wiki10-31K (You et al., 2019) | Sequential | 30,938 | 14,146 | - | 11,411 | 0.312 | 4.115 |
| ImageNet-LT (Liu et al., 2019) | Grid | 1,000 | 115,846 | - | 256 | 0.517 | 1.339 |
| Places-LT (Liu et al., 2019) | Grid | 365 | 62,500 | - | 996 | 0.610 | 2.387 |
| iNatural 2018 (Van Horn et al., 2018) | Grid | 8,142 | 437,513 | - | 500 | 0.647 | 1.658 |
| CIFAR 10-LT (100) (Cui et al., 2019) | Grid | 10 | 12,406 | - | 100 | 0.617 | 1.751 |
| CIFAR 10-LT (50) (Cui et al., 2019) | Grid | 10 | 13,996 | - | 50 | 0.593 | 1.751 |
| CIFAR 10-LT (10) (Cui et al., 2019) | Grid | 10 | 20,431 | - | 10 | 0.520 | 0.833 |
| CIFAR 100-LT (100) (Cui et al., 2019) | Grid | 100 | 10,847 | - | 100 | 0.498 | 1.972 |
| CIFAR 100-LT (50) (Cui et al., 2019) | Grid | 100 | 12,608 | - | 50 | 0.488 | 1.590 |
| CIFAR 100-LT (10) (Cui et al., 2019) | Grid | 100 | 19,573 | - | 10 | 0.447 | 0.836 |
| LVIS v0.5 (Gupta et al., 2019) | Grid | 1,231 | 693,958 | - | 26,148 | 0.381 | 6.250 |
| Cora-Full (Bojchevski & Günnemann, 2018) | Relational&Tabular | 70 | 19,793 | 146,635 | 62 | 0.321 | 0.919 |
| Wiki (Mernyei & Cangea, 2020) | Relational | 17 | 2,405 | 25,597 | 45 | 0.414 | 1.000 |
| Email (Yin et al., 2017) | Relational&Tabular | 42 | 1,005 | 25,934 | 109 | 0.413 | 1.263 |
| Amazon-Clothing (McAuley et al., 2015) | Relational&Tabular | 77 | 24,919 | 208,279 | 10 | 0.343 | 0.814 |
| Amazon-Electronics (McAuley et al., 2015) | Relational&Tabular | 167 | 42,318 | 129,430 | 9 | 0.329 | 0.600 |

CiteSeer (Giles et al., 1998), and Amazon (McAuley et al., 2015). Moreover, in knowledge graphs like YAGO (Suchanek et al., 2007) and DBpedia (Auer et al., 2007), the distribution of entities may exhibit long-tailed distribution, with only a few entities densely connected to others.

In HEROLT, we collect 16 datasets of various types (tabular, sequential, grid, and relational data) across 3 domains (natural language processing, computer vision, and graph mining) as discussed in Appx. B.3. Table 2 shows data statistics (*e.g.*, size, #categories) and long-tailedness of these datasets.

**Angle 3: Task Heterogeneity with 15 Algorithms on 5 Tasks.** Visual recognition has long been recognized as a significant aspect of the study of long-tailed problems. Beyond that, however, real-world applications require different tasks with unique learning objectives, presenting unique challenges for long-tailed algorithms. Despite its importance, this crucial angle has not been well explored in existing benchmarks. To fill this gap, we aim to benchmark long-tailed algorithms based on various tasks they are designed to solve.

- **Classification** (Fu et al., 2023; Jain & Kapoor, 2009) assigns each sample to one label. Data imbalance usually occurs in binary classification or multi-class classification, while long-tailed problems focus on multi-class classification with a large number of categories.

- **Multi-label text classification** (Chang et al., 2020; Zhang et al., 2021b; Yu et al., 2022; Liu et al., 2017) involves assigning the most relevant subset of class labels from an extremely large label set to each document. However, the extremely large label space often leads to significant data scarcity, particularly for rare labels in the tail. Consequently, many long-tailed algorithms are specifically designed to address the long-tailed problem in this task. Additionally, tasks such as sentence-level few-shot relation classification (Fan et al., 2019) and information extraction (containing three sub-tasks named relation extraction, entity recognition, and event detection) (Nan et al., 2021) are also frequently addressed by long-tailed algorithms.

- **Image classification** (Kang et al., 2020; Zhou et al., 2020; Tang et al., 2020; Zhong et al., 2021; Cao et al., 2020a; Zhong et al., 2019), which involves assigning a label to an entire image, is a widely studied task in long-tailed learning that received extensive research attention. Furthermore, there are some long-tailed algorithms focusing on similar tasks, including out-of-distribution detection (Wang et al., 2022; Bai et al., 2023), image retrieval (Long et al., 2022; Kou et al., 2022), image generation (He et al., 2020; Rangwani et al., 2022), visual relationship recognition (Wang et al., 2020b; Abdelkarim et al., 2021) and video classification (Li et al., 2019; Zhang et al., 2021d).

- **Instance segmentation** (Ren et al., 2020; Tan et al., 2020) is a common and crucial task that gained significant attention in the development of long-tailed algorithms aimed at enhancing the performance of tail classes. It involves the identification and separation of individual objects within an image, including the detection of object boundaries and the assignment of a unique label to each object. It contains several parts: Object detection (Pan et al., 2021; Wang et al., 2021b) and Semantic segmentation (Zhang et al., 2021c).

- **Node classification** (Zhao et al., 2021; Qu et al., 2021; Liu et al., 2021b; Yun et al., 2022) involves assigning labels to nodes in a graph based on node features and connections between them.

Table 3: Algorithms available in the HEROLT benchmark.

| Algorithm | Venue | Long-tailedness | Task |
|---|---|---|---|
| SMOTE (Chawla et al., 2002) | 02JAIR | Data imbalance | Classification |
| NearMiss (Mani & Zhang, 2003) | 03ICML | Data imbalance | Classification |
| X-Transformer (Chang et al., 2020) | 20KDD | Data imbalance, extreme # of categories | Multi-label text classification |
| XR-Transformer (Zhang et al., 2021b) | 21NeurIPS | Data imbalance, extreme # of categories | Multi-label text classification |
| XR-Linear (Yu et al., 2022) | 22KDD | Data imbalance, extreme # of categories | Multi-label text classification |
| BBN (Zhou et al., 2020) | 20CVPR | Data imbalance | Image classification |
| BALMS (Ren et al., 2020) | 20NeurIPS | Data imbalance | Image classification, Instance segmentation |
| OLTR (Liu et al., 2019) | 19CVPR | Data imbalance, extreme # of categories | Image classification |
| TDE (Tang et al., 2020) | 20NeurIPS | Data imbalance | Image classification |
| MiSLAS (Zhong et al., 2021) | 21CVPR | Data imbalance | Image classification |
| Decoupling (Kang et al., 2020) | 20ICLR | Data imbalance | Image classification |
| GraphSMOTE (Zhao et al., 2021) | 21WSDM | Data imbalance | Node classification |
| ImGAGN (Qu et al., 2021) | 21KDD | Data imbalance | Node classification |
| TailGNN (Liu et al., 2021b) | 21KDD | Data imbalance, extreme # of categories | Node classification |
| LTE4G (Yun et al., 2022) | 22CIKM | Data imbalance, extreme # of categories | Node classification |

Table 4: Comparing the methods on long-tailed tabular datasets. Each point is the mean and standard deviation of 10 runs. "Time" means training time plus inference time.

| Dataset | Method | Acc | bAcc | Precision | Recall | mAP | Time (s) |
|---|---|---|---|---|---|---|---|
| Glass | SMOTE | 97.0±1.1 | 98.5±0.5 | 95.1±1.1 | 98.5±0.5 | 99.9±0.0 | 0.6 |
| | NearMiss | 76.7±0.0 | 88.1±0.0 | 92.1±0.0 | 88.1±0.0 | 98.9±0.0 | 0.4 |
| Abalone | SMOTE | 20.1±1.1 | 19.1±1.1 | 11.5±0.6 | 19.1±1.1 | 17.4±0.2 | 12.4 |
| | NearMiss | 23.4±0.0 | 10.3±0.0 | 7.0±0.0 | 10.3±0.0 | 18.8±0.0 | 2.8 |

The emergence of long-tailed algorithms for this task is relatively recent, but the development is flourishing. Additionally, there are some long-tailed learning studies addressing entity alignment tasks (Zeng et al., 2020; Cao et al., 2020b) in knowledge graphs.

In HEROLT, we have a comprehensive collection of algorithms for classification, multi-label text classification, image classification, instance segmentation, and node classification tasks (Table 3). We discuss what technologies the algorithms use and how they solve long-tailed problems in Appx. B.2.

## 3 EXPERIMENT RESULTS AND ANALYSES

We conduct extensive experiments to further answer the question: how do long-tailed learning algorithms perform with regard to different tasks on different domains? In this section, we present the performance of 15 state-of-the-art algorithms on 5 typical long-tailed learning tasks across natural language processing, computer vision, and graph mining.

### 3.1 EXPERIMENT SETTING

**Hyperparameter Settings.** For all the 15 algorithms in HEROLT, we use their default hyperparameter settings in the original paper for a fair comparison. Refer to the Appx. C.1 for more information. **Evaluation Metrics.** We evaluate different long-tailed algorithms by several basic metrics, which are divided into accuracy-based metrics, including accuracy (Acc) (Sorower, 2010), precision (Grandini et al., 2020), recall (Grandini et al., 2020), and balanced accuracy (bAcc) (Grandini et al., 2020); ranking-based metrics such as mean average precision (MAP) (Rezatofighi et al., 2019); and running time. The computation formula and description of the metrics are outlined in Table 9 in Appx. C.2.

### 3.2 ALGORITHM PERFORMANCE ON CLASSIFICATION

Recently, there has been very limited work considering classification tasks on pure tabular long-tailed data. We compare the performance of two methods in Table 4. We find: **(1)** SMOTE (using an upsampling technique) shows superior performance to NearMiss (using a downsampling technique). Specifically, SMOTE is 85.43% highly balanced accuracy than NearMiss on the Abalone dataset. **(2)** While Acc treats all samples equally, bAcc considers data imbalance by averaging across classes. Although NearMiss achieves higher accuracy on the imbalanced Abalone dataset, it tends to exhibit a bias toward majority classes and does not sufficiently improve the minority classes, resulting in a lower bAcc score. Precision provides an insight into how much we can trust the model when it identifies a sample as Positive. Precision of NearMiss is lower as it may wrongly classify minority samples into other classes. Recall assesses the model's ability to identify all the Positive samples.

Table 5: Comparison of different methods on long-tailed sequential datasets for multi-label text classification tasks. "Time" refers to inference time. The results of bAcc@k are not reported since no relevant literature discusses how to use this metric in the multi-label setting.

| Dataset | Method | Acc | | | Precision | | | Recall | | | MAP | | | Time (s) |
|---|---|---|---|---|---|---|---|---|---|---|---|---|---|---|
| | | @1 | @3 | @5 | @1 | @3 | @5 | @1 | @3 | @5 | @1 | @3 | @5 | |
| Eurlex-4K | XR-Transformer | **88.2** | **75.8** | 62.8 | **88.2** | **75.8** | 62.8 | **17.9** | **45.2** | 61.1 | **88.2** | **82.0** | **75.6** | 66.9 |
| | X-Transformer | 87.0 | 75.2 | **62.9** | 87.0 | 75.2 | **62.9** | 17.7 | 44.8 | **61.2** | 87.0 | 81.1 | 75.1 | 433.9 |
| | XR-Linear | 82.1 | 69.6 | 58.2 | 82.1 | 69.6 | 58.2 | 16.6 | 41.4 | 56.6 | 82.1 | 75.9 | 69.9 | **0.2** |
| AmazonCat-13K | XR-Transformer | **96.7** | 83.6 | 67.9 | **96.7** | 83.6 | 67.9 | **27.7** | 63.3 | 79.0 | **96.7** | 90.5 | 83.1 | 78.3 |
| | X-Transformer | **96.7** | **83.9** | **68.6** | **96.7** | **83.9** | **68.6** | 27.6 | **63.4** | **79.7** | **96.7** | **90.6** | **83.4** | 428.6 |
| | XR-Linear | 93.0 | 78.9 | 64.3 | 93.0 | 78.9 | 64.3 | 26.3 | 59.7 | 75.2 | 93.0 | 86.1 | 78.9 | **0.3** |
| Wiki10-31K | XR-Transformer | 88.0 | **79.5** | **69.7** | 88.0 | **79.5** | **69.7** | **5.3** | **14.0** | **20.1** | 88.0 | **83.9** | **79.2** | 117.3 |
| | X-Transformer | **88.5** | 78.5 | 69.1 | **88.5** | 78.5 | 69.1 | **5.3** | 13.8 | 19.8 | **88.5** | 83.6 | 78.7 | 433.2 |
| | XR-Linear | 84.6 | 73.0 | 64.3 | 84.6 | 73.0 | 64.3 | 5.0 | 12.7 | 18.4 | 84.6 | 78.7 | 73.7 | **1.1** |

Table 6: Comparison of different methods on natural long-tailed grid datasets. "Time" refers to inference time. Recall equals to bAcc for multi-class classification.

| Dataset | Task | Method | Acc | | | | Precision | Recall | MAP | Time (s) |
|---|---|---|---|---|---|---|---|---|---|---|
| | | | Many | Medium | Few | Overall | | (bAcc) | | |
| ImageNet-LT | Image classification | BBN | 59.4 | 45.4 | 16.3 | 46.6 | 47.5 | 46.6 | 24.9 | 43.0 |
| | | BALMS | 50.1 | 39.6 | 25.3 | 41.6 | 41.2 | 41.6 | 20.6 | 29.0 |
| | | OLTR | 46.0 | 36.9 | 18.4 | 37.8 | 37.2 | 7.8 | 16.9 | 36.9 |
| | | TDE | **61.6** | **47.2** | 30.5 | **50.5** | **50.5** | 50.8 | **29.0** | 30.9 |
| | | MiSLAS | 60.9 | 46.8 | **32.5** | 50.0 | 39.0 | 38.0 | 21.0 | **18.9** |
| | | Decoupling | 58.4 | 43.7 | 28.2 | 47.4 | 35.2 | 36.1 | 19.4 | 21.8 |
| Places-LT | Image classification | BBN | 34.8 | 32.0 | 5.8 | 27.5 | 30.7 | 27.5 | 9.4 | **15.4** |
| | | BALMS | 41.0 | 39.9 | 30.2 | 38.3 | 38.1 | 38.3 | 17.1 | 29.0 |
| | | OLTR | **44.0** | 40.6 | 28.5 | 39.3 | 39.5 | 39.3 | 17.9 | 30.6 |
| | | TDE | 33.9 | 27.8 | 19.3 | 28.4 | 28.4 | 29.7 | 10.4 | 18.6 |
| | | MiSLAS | 42.4 | **41.8** | **34.7** | **40.5** | **41.0** | **40.5** | **19.2** | 16.2 |
| | | Decoupling | 40.6 | 39.1 | 28.6 | 37.6 | 37.6 | 38.2 | 16.7 | 28.9 |
| iNatural 2018 | Image classification | BBN | 61.8 | **73.5** | 67.7 | 69.7 | 72.8 | 69.7 | 55.9 | 29.3 |
| | | BALMS | 61.8 | 64.6 | 66.4 | 65.1 | 68.1 | 65.1 | 50.2 | 24.1 |
| | | OLTR | 67.9 | 50.7 | 37.8 | 46.8 | 51.4 | 46.8 | 33.2 | 32.5 |
| | | TDE | 67.8 | 63.9 | 55.2 | 60.9 | 60.9 | 63.2 | 45.4 | 24.8 |
| | | MiSLAS | 72.3 | 72.3 | 70.7 | 71.6 | **74.6** | 71.6 | 58.0 | **19.6** |
| | | Decoupling | **74.3** | 72.4 | **71.2** | **72.1** | 72.1 | **75.4** | **58.9** | 23.9 |
| LVIS v0.5 | Instance segmentation | BALMS | 62.9 | 34.7 | 16.1 | 60.0 | 37.1 | 46.8 | 37.1 | 1436.1 |

NearMiss fails to characterize minority classes, it typically scores lower on Recall. Similarly, MAP is calculated by averaging across all classes and exhibits a similar trend with the other metrics.

### 3.3 ALGORITHM PERFORMANCE ON MULTI-LABEL TEXT CLASSIFICATION

In Table 5, we provide a performance comparison of three methods for multi-label text classification tasks on sequential datasets. We find: **(1)** Among these methods, XR-Transformer and X-Transformer demonstrate comparably superior performance on multiple metrics. On Eurlex-4K, XR-Transformer achieves a 1.37% improvement in ACC@1 compared to the second-best method. On AmazonCat-13K, X-Transformer exhibits a 1.03% improvement in Acc@5 than the suboptimal method. **(2)** Conversely, XR-Linear consistently exhibits the poorest performance across all three datasets, which verifies the effectiveness of recursive learning of transformer encoders than linear methods. However, XR-Linear is more than 100x faster than XR-Transformer, making it suitable for scenarios with large dataset sizes and strict requirements on time-consuming. **(3)** Notably, all methods exhibit a limitation in effectively recognizing certain classes, as indicated by the observed low recalls across all datasets.

### 3.4 ALGORITHM PERFORMANCE ON IMAGE CLASSIFICATION AND INSTANCE SEGMENTATION

Among the considered algorithms, BBN and MisLAS employ mixup techniques; BALMS and OLTR utilize meta-learning; TDE uses causal inference; and MisLAS, Decoupling, and BALMS decouple the learning process. We have the following observations from the results in Table 6: **(1)** No single technique (*e.g.*, meta-learning, decoupled training, or mixup) can consistently perform the

Table 7: Comparing the methods on semi-synthetic long-tailed datasets with three imbalance factors (100, 50, 10). "Time" refers to inference time. Recall equals to bAcc for multi-class classification.

| Dataset | Method | Acc | | | Precision | | | Recall (bAcc) | | | MAP | | | Time (s) |
|---|---|---|---|---|---|---|---|---|---|---|---|---|---|---|
| | | 100 | 50 | 10 | 100 | 50 | 10 | 100 | 50 | 10 | 100 | 50 | 10 | |
| CIFAR-10-LT | BBN | 80.7 | 83.4 | 88.7 | 81.0 | 84.0 | 88.8 | 80.7 | 83.4 | 88.7 | 67.5 | 71.8 | 80.0 | 5.2 |
| | BALMS | 84.2 | 86.7 | 91.5 | 84.3 | 86.7 | 91.5 | 84.2 | 86.7 | 91.5 | 72.8 | 76.9 | 84.8 | **1.7** |
| | OLTR | **88.7** | **90.8** | **93.3** | **90.3** | **91.6** | **93.5** | **88.7** | **90.8** | **93.3** | **80.6** | **83.8** | **87.8** | 9.3 |
| | TDE | 80.9 | 82.9 | 88.3 | 80.9 | 82.9 | 88.3 | 81.4 | 83 | 88.2 | 68.0 | 70.7 | 79.2 | 1.8 |
| | MiSLAS | 82.5 | 85.7 | 90.0 | 83.4 | 86.1 | 90.1 | 82.5 | 85.7 | 90.1 | 70.5 | 75.2 | 82.3 | 8.2 |
| | Decoupling | 53.5 | 61.2 | 73.4 | 53.5 | 61.2 | 73.4 | 55.4 | 62.3 | 73.5 | 34.0 | 42.0 | 57.3 | 1.8 |
| CIFAR-100-LT | BBN | 40.9 | 46.7 | 59.7 | 43.3 | 48.0 | 59.9 | 40.9 | 46.7 | 59.7 | 19.9 | 24.5 | 38.2 | 4.3 |
| | BALMS | 51.0 | 56.2 | 55.2 | 50.0 | 56.0 | 54.8 | 51.0 | 56.2 | 55.2 | 29.5 | 34.9 | 33.6 | 2.2 |
| | OLTR | **67.6** | **72.0** | **77.2** | **74.0** | **77.2** | **79.0** | **67.6** | **72.0** | **77.2** | **49.8** | **55.2** | **61.6** | 9.3 |
| | TDE | 43.5 | 48.9 | 58.9 | 43.5 | 48.9 | 58.9 | 42.8 | 49.3 | 59.7 | 21.0 | 26.0 | 36.8 | **1.7** |
| | MiSLAS | 47.0 | 52.3 | 63.3 | 46.5 | 52.5 | 63.4 | 47.0 | 52.3 | 63.2 | 25.4 | 30.5 | 42.3 | 7.2 |
| | Decoupling | 34.0 | 36.4 | 51.5 | 33.9 | 36.4 | 51.5 | 32.2 | 35.7 | 51.4 | 14.0 | 15.8 | 29.2 | 1.8 |

best across all tasks and datasets, and there are limited algorithms that consider both classification and segmentation tasks. Therefore, in contrast to taxonomy based on techniques in methods, the three novel angles we propose may be more suitable for benchmarking. **(2)** MisLAS consistently performs top two best in accuracy on four natural long-tailed datasets. In particular, MisLAS exhibits a remarkable overall accuracy of 40.5% on Places-LT dataset, surpassing the suboptimal method by 4.65%. The superiority of MisLAS is further evident in the tail category, where its few-shot accuracy surpasses other methods by 9.81% on Places-LT. **(3)** The few-shot accuracy is often lower than the many-shot accuracy for all methods. For example, on Image-LT and Places-LT, BBN exhibits particularly poor performance in terms of few-shot accuracy, with a decrease of 72.56% and 83.33% respectively compared to many-shot accuracy. But on iNatural 2018, BALMS achieves a few-shot accuracy of 66.4%, surpassing many-shot accuracy of 61.7% and medium-shot accuracy of 64.8%.

CIFAR-10-LT and CIFAR-100-LT are semi-synthetic datasets, where the number of samples in each class is determined by a controllable imbalance factor (typically 10, 50, 100). Similarly, we have the observations as shown in Table 7: **(1)** OLTR and BALMS, two decoupled learning-based method, shows the best and second-best accuracy performance on the synthetic CIFAR-10-LT and CIFAR-100-LT with varying degrees of IF, suggesting that decoupled learning may have potential in handling long-tailed problems. **(2)** As we control IF of the synthetic datasets to be larger, the long-tailed phenomenon is more severe (showing higher values of Gini coefficient, IF, and Pareto-LT Ratio), and the performance of nearly all methods exhibits a decline. **(3)** CIFAR-100-LT dataset appears more affected than CIFAR-10-LT under the different settings of IF, possibly because it has a more severe long-tailed phenomenon with a larger number of categories. **(4)** Compared to natural datasets, semi-synthetic datasets exhibit lower IFs and Gini coefficients but higher Pareto-LT Ratios. Following the guideline provided in *Angle 1*, this observation may indicate the need for increased focus on addressing the challenge caused by the number of categories for synthetic datasets (but it does not mean synthetic datasets necessarily contain more categories than real datasets). The insight may elucidate why OLTR method, which can characterize the extensive number of categories, exhibits more significant performance improvements on synthetic datasets than real datasets.

## 3.5 ALGORITHM PERFORMANCE ON NODE CLASSIFICATION

From the experimental results of various long-tailed algorithms for node classification on relational datasets in Table 8, we have: **(1)** No method consistently outperforms all others across all datasets. On the same dataset, the performance of different runs may have large variances, *e.g.*, LTE4G on Wiki. **(2)** It is worth noting that GraphSmote exhibits promising performance on Wiki dataset. Wiki presents high IF and Gini coefficient yet a low Pareto-LT Ratio, hinting that the main challenge may stem from data imbalance as discussed in *Angle 1*. Therefore, employing the simple augmentation method may yield great results. **(3)** Tail-GNN exhibits superior performance on Cora-full and Amazon_clothing datasets. Especially on Cora-full, Tail-GNN achieves a 4.93% higher accuracy than the second-best method. However, the scalability of Tail-GNN shows limitations. It faces out-of-memory problems on Amazon_Eletronics with 42,318 nodes and 129,430 edges. **(4)** The performance of ImGAGN is relatively weak since it considers only one class as the tail class by default. This limitation becomes apparent in datasets with a large number of classes. Nonetheless, ImGAGN shows a performance improvement by adjusting the number of classes considered as tail. In addition, ImGAGN is less time-consuming and is 5x faster than LTE4G on the largest Amazon_Eletronics dataset.

Table 8: Comparing the methods on long-tailed relational datasets. Each point is the mean and standard deviation of 10 runs. "Time" means training time plus inference time.

| Dataset | Method | Acc | bAcc | Precision | Recall | MAP | Time (s) |
|---|---|---|---|---|---|---|---|
| Cora-Full | GraphSmote | 60.5±0.8 | 51.9±0.6 | 60.2±0.8 | 51.9±0.6 | 54.9±0.4 | 718.8 |
| | ImGAGN | 4.2±0.8 | 1.5±0.1 | 0.2±0.1 | 1.5±0.1 | 2.7±0.1 | **69.6** |
| | Tail-GNN | **63.8±0.3** | **54.6±0.6** | **63.8±0.4** | **54.6±0.6** | **66.8±0.6** | 906.2 |
| | LTE4G | 60.8±0.5 | **54.6±0.5** | 61.5±0.6 | **54.6±0.5** | 51.1±0.9 | 281.6 |
| Wiki | GraphSmote | **66.0±0.9** | **51.1±2.0** | **66.3±1.1** | **51.1±2.0** | 64.4±1.9 | 52.3 |
| | ImGAGN | 45.4±6.7 | 24.5±4.1 | 44.8±5.0 | 24.5±4.1 | 64.1±6.2 | **14.3** |
| | Tail-GNN | 63.6±0.7 | 47.7±1.1 | 64.0±1.4 | 47.7±1.1 | **65.0±2.1** | 22.5 |
| | LTE4G | 58.2±18.5 | 48.9±12.5 | 60.2±20.1 | 48.9±12.5 | 59.3±16.5 | 53.1 |
| Email | GraphSmote | 58.3±1.2 | 34.2±1.4 | 54.4±2.0 | 34.2±1.4 | 44.6±2.4 | 126.2 |
| | ImGAGN | 42.7±1.9 | 23.0±1.4 | 38.4±2.5 | 23.0±1.4 | 35.7±2.2 | 19.7 |
| | Tail-GNN | 56.5±1.7 | **34.5±1.6** | **55.6±0.4** | **34.5±1.6** | **58.0±3.0** | **8.6** |
| | LTE4G | **58.7±2.0** | 34.3±2.1 | 54.1±2.5 | 34.3±2.1 | 47.0±2.1 | 72.1 |
| Amazon_Clothing | GraphSmote | 66.3±0.4 | 63.2±0.4 | 64.9±0.3 | 63.2±0.4 | 57.6±0.9 | 919.1 |
| | ImGAGN | 30.3±1.1 | 12.8±0.7 | 23.3±1.2 | 12.8±0.7 | 32.0±0.7 | **89.6** |
| | Tail-GNN | **69.2±0.6** | **65.9±0.5** | **67.7±0.5** | **65.9±0.5** | **68.7±0.1** | 768.7 |
| | LTE4G | 65.6±0.6 | 64.2±0.6 | 64.8±0.5 | 64.2±0.6 | 54.3±2.7 | 349.7 |
| Amazon_Eletronics | GraphSmote | **56.2±0.5** | 51.7±0.5 | 55.6±0.4 | 51.7±0.5 | **36.2±1.9** | 3406.2 |
| | ImGAGN | 17.1±1.1 | 7.4±0.6 | 12.3±0.7 | 7.4±0.6 | 16.8±0.7 | **218.5** |
| | Tail-GNN | OOM | OOM | OOM | OOM | OOM | OOM |
| | LTE4G | 55.8±0.4 | **53.0±0.5** | **56.1±0.3** | **53.0±0.5** | 32.7±2.2 | 1112.8 |

## 4 RELATED WORK

Recently, several papers that review long-tailed problems have been published. One of the earliest works (Zhang et al., 2021f) aims to improve the performance of long-tailed algorithms through a reasonable combination of existing tricks. Yang et al. (2022) conducts a survey on long-tailed visual recognition and shows a taxonomy of existing methods. Fu et al. (2022) divides methods into three categories, namely, training, fine-tuning, and inference stage. Fang et al. (2023) groups methods based on balanced data, balanced feature representation, balanced loss, and balanced prediction. Zhang et al. (2023) categorizes them into class re-balancing, information augmentation, and module improvement. Although these papers summarize studies on long-tailed visual recognition, they pay less attention to the extent of long-tailedness and fail to consider data complexity and task heterogeneity.

Next, we highlight the similarities and differences of long-tailed learning with related areas, e.g., imbalanced learning and few-shot learning. Imbalanced Learning focuses on learning from imbalanced data, where minority classes contain significantly fewer training samples than majority classes. In imbalance learning, the class numbers may be small, while the number of minority samples is not necessarily small. But in long-tailed learning, the number of classes is larger and samples in tail classes are often very scarce. Few-shot Learning aims to train well-performing models from limited supervised samples. Long-tailed datasets, like few-shot datasets, have limited labeled samples in tail classes but with an imbalanced distribution. In contrast, few-shot datasets tend to be more balanced.

## 5 CONCLUSIONS AND FUTURE WORK

In this paper, we introduce HEROLT, the most comprehensive heterogeneous long-tailed learning benchmark with 15 state-of-the-art algorithms and 6 evaluation metrics on 16 real-world benchmark datasets across 5 tasks from 3 domains. Based on the analyses of three pivotal angles, we gain valuable insights into the characterization of data long-tailedness, the data complexity of various domains, and the heterogeneity of emerging tasks. Our benchmark and evaluations are released at `https://anonymous.4open.science/r/HeroLT-9746/`.

On top of them, we suggest intellectual challenges and promising research directions in long-tailed learning: **(C1)** Theoretic Challenge: Current work lacks sufficient theoretical tools for analyzing long-tailed models, such as their generalization performance. **(C2)** Algorithmic Challenge: Existing research typically focuses on one task in one domain, while there is a trend to consider multiple forms of input data (e.g., text and images) by multi-modal learning (Guo et al., 2019; Baltrušaitis et al., 2018; Zhang et al., 2020), or to solve multiple learning tasks (e.g., segmentation and classification) by multi-task learning (Ruder, 2017; Crawshaw, 2020). **(C3)** Application Challenge: In open environments, many datasets exhibit long-tailed distributions. However, long-tailed problems in domains like antibiotic resistance genes (Arango-Argoty et al., 2018; Li et al., 2021b) receive insufficient attention.

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

# A LONG-TAILED DATA DISTRIBUTIONS

In this section, we will give some real-world examples with long-tailed distributions. Cora-Full dataset consists of 19,793 scientific publications classified into one of seventy categories (Bojchevski & Günnemann, 2018). As shown in Fig. 3(a), this dataset exhibits a prominent long-tailed distribution, wherein the number of instances belonging to the head categories far surpasses that of the tail categories. Similarly, Amazon_Eletronics dataset (McAuley et al., 2015) also exhibits long-tailed distribution (see Fig. 3(b)), where each product is considered as a node belonging to a product category in "Electronics." Despite the emergence of machine learning methods aimed at facilitating accurate classification, further solutions are called due to the challenges of long-tailed distribution.

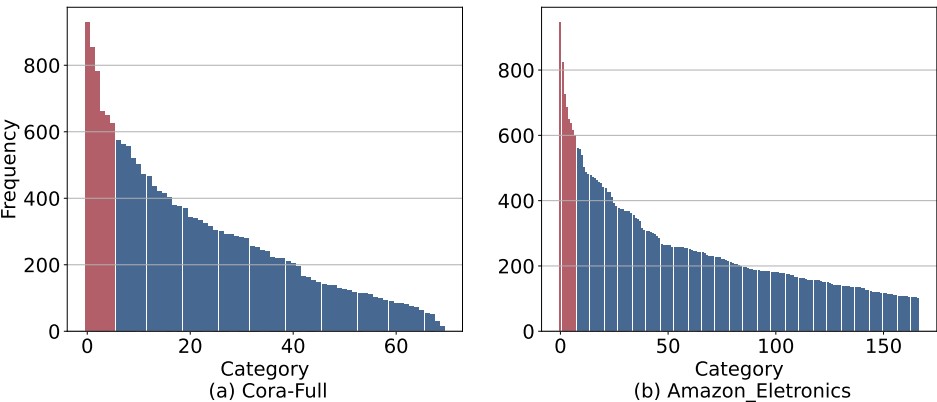

Figure 3: The data distributions on two commonly used datasets exhibit prominent long-tailed distributions.

In addition, we present a motivative application of the recommendation system (Liu et al., 2022) as shown in Fig. 4, which naturally exhibits long-tailed data distributions coupled with data complexity [2] (e.g., tabular data and relational data) and task heterogeneity (e.g., user profiling [1] and recommendation [2]). Additionally, heterogeneous long-tailed learning has various real-world applications, such as financial fraud detection (Reurink, 2019; Karpoff, 2021) and ARG prediction (Weinstock et al., 2003).

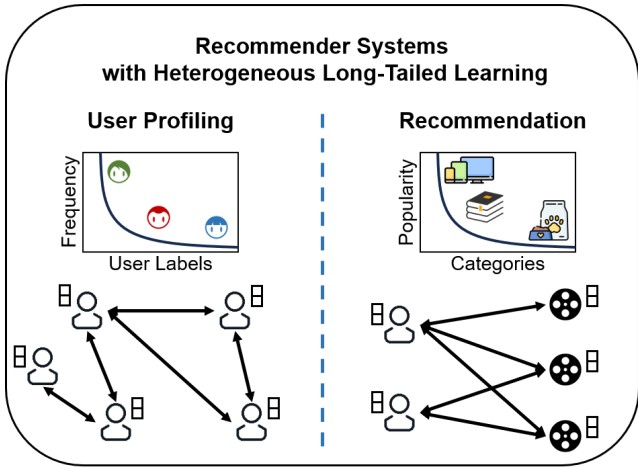

Figure 4: The data distributions on two commonly used datasets exhibit prominent long-tailed distributions.

## B    MORE DETAILS ON HEROLT

### B.1    LONG-TAILEDNESS METRICS

To measure the long-tailedness of a dataset, a commonly used metric is the imbalance factor, and (Yang et al., 2022) propose using Gini coefficient as a measurement. In this section, we analyze the strengths and weaknesses of each metric and suggest using Pareto-LT Ratio to evaluate the long-tailedness of a dataset.

**Imbalance Factor.** To measure the skewness of long-tailed distribution, (Cui et al., 2019) first introduces the imbalance factor as the size of the largest majority class to the size of the smallest minority class:

$$IF = n_1/n_C \tag{1}$$

where $n_c, c = 1, 2, \ldots, C$, represents the size of each category following descending order. The range of imbalance factor is $[1, \infty)$. Intuitively, the larger the value of IF indicates a more imbalanced dataset.

**Gini Coefficient.** Gini coefficient is a measure of income inequality used to quantify the extent to which the distribution of income among a population deviates from perfect equality. (Yang et al., 2022) propose to use Gini coefficient as a long-tailedness metric since long-tailedness is similar to inequality between each category.

$$Gini = \frac{\sum_{i=1}^{C} \sum_{j=1}^{C} |n_i - n_j|}{2nC} \tag{2}$$

Gini coefficient ranges from 0 to 1, where a larger value indicates that the dataset is more imbalanced.

**Pareto-LT Ratio.** We propose a new metric named Pareto-LT Ratio to measure the long-tailedness. The design of this metric is inspired by the Pareto distribution, which is defined as:

$$Pareto - LT = \frac{C - Q(0.8)}{Q(0.8)} \tag{3}$$

where $Q(p) = min\{y : Pr(\mathcal{Y} \leq y) = p, 1 \leq y \leq T\}$ is the quantile function of order $p \in (0, 1)$ for $\mathcal{Y}$. The numerator represents the number of categories to which the last 20% instances belongs, and the denominator represents the number of categories to which the other 80% instances belongs in the dataset. Intuitively, the higher the skewness of the data distribution, the larger the ratio will be; the more classes, the larger the long-tailedness ratio. In addition to serving as a valuable guideline for model selection, Pareto-LT Ratio has the potential to analyze the generalization performance of long-tailed learning. According to Maurer et al. (2016), we obtain new generalization bounds by measuring the complexity of classifying long-tailed data in terms of Pareto-LT Ratio.

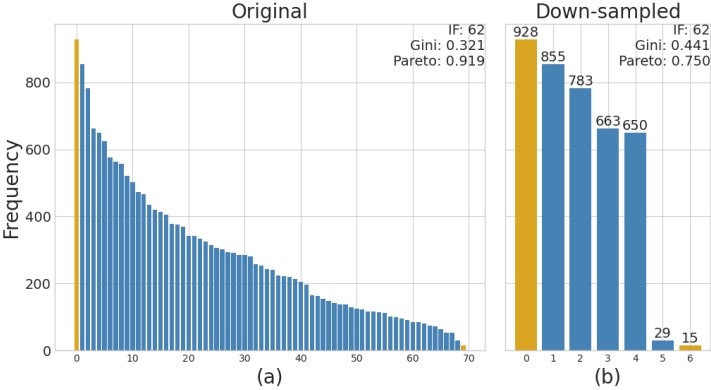

Figure 5: An example to compare the three long-tailedness metrics.

Imbalance factor is intuitive and easy to calculate, but it only considers the imbalance between the largest and smallest classes. Gini coefficient indicates the overall degree of category imbalance,

unaffected by extreme samples or absolute data size. However, the two metrics pay more attention to data imbalance and may not reflect the number of categories. Our proposed Pareto-LT Ratio characterizes two properties of long-tailed datasets: (1) data imbalance and (2) extreme # of categories. For better understanding, we give a specific example of the three long-tailedness metrics (as shown in Figure 5). As the number of categories increases, the difficulty of classifying a long-tailed dataset therefore increases. For example, we down-sampled 7 categories from the original Cora-Full dataset. Although the two datasets are clearly different, the imbalance factor remains the same (62 for both the original and down-sampled datasets) as the number of samples in the most majority and most minority categories does not change. Gini coefficient indicates the overall degree of category imbalance and thus shows a large increase of value in the down-sampled dataset (0.321 for original and 0.441 for down-sampled). However, the data complexity also includes the number of categories, i.e., 70 classes in Fig(a) v.s 7 classes in Fig(b), which is not reflected in imbalance factor and Gini coefficient. As the number of categories of the down-sampled dataset decreases dramatically, Pareto-LT Ratio better characterizes the differences between the original Cora dataset (0.919) and its down-sampled dataset (0.750) by a small decrease of the metric value.

## B.2 HeroLT Algorithm List

In this section, we introduce 15 popular and recent methods for solving long-tailed problems in our benchmark according to whether they can solve the problems of data imbalance and an extreme number of categories. The baseline algorithms are selected due to the following reasons: (1) Data Complexity: The selected methods provide comprehensive coverage of heterogeneous data (i.e, tabular, sequential, grid, and relational data) in long-tailed learning problems. (2) Task Heterogeneity: we explore a range of techniques for long-tailed learning approaches (i.e., data augmentation, meta-learning, decoupled training, mixup), designing for various tasks (i.e, classification, multi-label text classification, image classification, and node classification). (3) SOTA Performance: Most of the selected methods are the SOTAs, which are recently published and highly cited. (4) Open Source: All of the selected methods are open-sourced in GitHub. In addition, our toolbox is still being updated, and we will include more algorithms in the future.

**SMOTE** (Chawla et al., 2002) generates synthetic samples by feature interpolation minority samples with their nearest.

**NearMiss** (Mani & Zhang, 2003) is a downsampling method that selects samples based on the distance of majority class samples to minority class samples.

**X-Transformer** (Chang et al., 2020) consists of three components: semantic label indexing, deep neural matching, and ensemble ranking. Semantic label indexing decomposes extreme multi-label classification (XMC) problems into a feasible set of subproblems with much smaller output space by label clustering; deep neural matching fine-tunes a Transformer model for each subproblem; ensemble ranking is conditionally trained based on the instance-cluster assignment and embeddings from the Transformer and is used to aggregate scores from subproblems. X-Transformer solves data imbalance and an extreme number of categories.

**XR-Transformer** (Zhang et al., 2021b) is a transformer-based XMC framework that fine-tunes pre-trained transformers recursively leveraging multi-resolution objectives and cost-sensitive learning. The embedding generated at the previous task is used to bootstrap the non pre-trained part for the current task. XR-Transformer solves data imbalance and an extreme number of categories.

**XR-Linear** (Yu et al., 2022) is designed for XMC problem. It consists of recursive linear models that traverse an input from the root of a hierarchical label tree to a few leaf node clusters and return top-k relevant labels within the clusters as predictions. XR-Linear solves data imbalance and an extreme number of categories.

**BBN** (Zhou et al., 2020) consists of two branches: the conventional learning branch with a uniform sampler for learning universal patterns for recognition and the re-balancing branch with a reversed sampler for modeling the tail data. Then the predicted outputs of these bilateral branches are aggregated in the cumulative learning part using an adaptive trade-off parameter, which first learns the universal features from the original distribution and then gradually pays attention to the tail data. By different data samplers (such as mixup which is popular for solving long-tailed problems) and cumulative learning strategy, BBN can address data imbalance.

**BALMS** (Ren et al., 2020) presents Balanced Softmax, an unbiased extension of Softmax, to accommodate the label distribution shift between training and testing. In addition, it applies a Meta Sampler to learn the optimal class re-sample rate by meta learning. Therefore, BALMS can address data imbalance.

**OLTR** (Liu et al., 2019) learns from natural long-tailed open-end distributed data. It consists of two main modules: dynamic meta embedding and modulated attention. The former combines a direct image feature and an associated memory feature to transfer knowledge between head and tail classes, and the latter maintains discrimination between them. Therefore, OLTR solves the challenges of data imbalance and an extreme number of categories.

**TDE** (Tang et al., 2020) is a framework for considering long-tailed problems using causal inference. It constructs a casual graph with four variables: momentum, object feature, projection of head direction, and model prediction, using casual intervention in training and counterfactual reasoning in inference to preserve "good" feature while cut off "bad" confounding effect. TDE solves data imbalance.

**MiSLAS** (Zhong et al., 2021) decouples representation learning and classifier learning. It uses mixup and designs label-aware smoothing to handle different degrees of over-confidence for classes and improve classifier learning. It also uses shift learning on the batch normalization layer to reduce dataset bias in the decoupling framework. MiSLAS can solve data imbalance.

**Decoupling** (Kang et al., 2020) decouples the learning procedure into representation learning and classification. The authors find that instance-balance sampling gives more generalizable representations and can achieve state-of-art performance after properly adjusting the classifiers. Decoupling can address the challenge of data imbalance.

**DisAlign** (Zhang et al., 2021c) develop a unified two-stage learning framework. This method introduces an adaptive calibration strategy for each sample to tackle the biased label prediction. The authors also develop a generalized re-weighting technique to utilize the category prior. The authors provide the experimental results in diverse visual recognition tasks including image classification, semantic segmentation, and object detection.

**PaCo** (Cui et al., 2021) presents Parametric Contrastive Learning. To mitigate the problem of contrastive loss bias on head categories from an optimization perspective, the authors propose a set of parametric class-wise learnable centers, which adaptively change the intensity of pushing samples belonging to the same category close to each other.

**GPaCo** (Cui et al., 2023) is a follow-up work of PaCo by removing the momentum encoder, which achieves better model performance and robustness.

**FCC** (Li et al., 2023) focuses on the effect of the density of Backbone Features on long-tail learning. It utilizes a scaling factor to compress features to force denser clusters during the training phase while keeping the original features during the testing phase.

**LTR** (Alshammari et al., 2022) emphasizes the importance of learning balanced weights via parameter regularization, including L2-normalization, weight decay, and MaxNorm regularizers. The study illustrates that weight decay and MaxNorm benefit learning balanced weights and improve accuracy. Therefore, the authors propose a two-stage training method based on these techniques.

**GraphSMOTE** (Zhao et al., 2021) is the first work considering node class imbalance on graphs proposed in 2021. It first uses a GNN-based feature extractor to learn node representations and then applies SMOTE to generate synthetic nodes for the minority classes. Then an edge generator pre-trained on the original graph is introduced to model the existence of edges among nodes. The augmented graph is used for classification with a GNN classifier. By generating nodes for minority classes, GraphSMOTE increases the number of labeled samples for these classes, thus addressing the data imbalance.

**ImGAGN** (Qu et al., 2021) is an adversarial learning-based approach. It uses a generator to generate a set of synthetic minority nodes and topological structures, then uses a discriminator to discriminate between real and fake (i.e., generated) nodes and between minority and majority nodes, which can solve the challenges of data imbalance. However, ImGAGN sets the smallest class as the minority class and the residual classes as the majority class, which fails when the dataset contains the large number of categories.

**Tail-GNN** (Liu et al., 2021b) Due to the specificity of rational data, the long-tailed problem on graphs includes the long-tailed of category and the long-tailed of node degree. Tail-GNN (Liu et al., 2021b) focuses on solving the long-tailed problem of degree by introducing transferable neighborhood translation to capture the relational tie between a node and its neighboring nodes. Then it complements missing information of the tail nodes for neighborhood aggregation. Tail-GNN learns robust node embeddings by narrowing the gap between head and tail nodes in terms of degree and addresses the challenges of data imbalance and an extreme number of categories in degree level.

**LTE4G** (Yun et al., 2022) splits the nodes into four balanced subsets considering class and degree long-tailed distributions. Then, it trains an expert for each balanced subset and employs knowledge distillation to obtain the head student and tail student for further classification. Lastly, LTE4G devises a class prototype-based inference. Because LTE4G uses knowledge distillation across the head and tail and considers the tail classes together, it can solve data imbalance and an extreme number of categories.

### B.3 HEROLT DATASET LIST

long-tailed challenges exist for various real-world data, such as tabular data, sequential data, grid data, and rational data. In this section, we briefly describe the collection of datasets selected for the initial version of our benchmark. In addition, we will show more datasets in past long-tailed studies on our toolbox page, including the statistics information (*e.g.*, size, #categories) and the long-tailedness (*e.g.*, imbalance factor, Gini coefficient, and Pareto-LT Ratio).

**Glass** (German, 1987) is a dataset from USA Forensic Science Service. Motivated by criminological investigation, the glass left at the scene of the crime can be classified into 6 types based on its oxide content.

**Abalone** (Nash & Ford, 1995) is a dataset for predicting the age of abalone from physical measurements, such as length, diameter, height, and weight.

**EURLEX-4K** (You et al., 2019) consists of legal documents from the European Union, and the number of instances in the training and test sets are 15,499 and 3,865.

**AMAZONCat-13K** (You et al., 2019) contains product descriptions from Amazon, and the number of instances in the training and test sets are 1,186,239 and 306,782.

**Wiki10-31K** (You et al., 2019) is a collection of Wikipedia articles, and the number of instances in the training and test sets are 14,146 and 6,616.

**ImageNet-LT** (Liu et al., 2019) is a long-tailed version sampled from the original ImageNet-2012, which is a large-scale image dataset constructed based on the WorldNet structure. The train set has 115,846 images from 1000 categories with maximally 1280 images per class and minimally 5 images per class. The test and valid sets have 50,000 and 20,000 samples and are balanced.

**Places-LT** (Liu et al., 2019) is a long-tailed version of scene classification dataset Places-2. The train set contains 62,500 images from 365 categories with maximally 4980 images per class and minimally 5 images per class. The test and valid sets are balanced with 100 and 20 images per class accordingly.

**iNatural 2018** (Van Horn et al., 2018) is a species classification dataset with a train set with 437,513 images for 8142 classes. The class frequencies follow a natural power law distribution with a maximum number of 4,980 images per class and a minimum number of 5 images per class. The test and valid sets contain 149,394 and 24,426 images, respectively.

**CIFAR 10-LT** and **CIFAR 100-LT** (Cui et al., 2019). The original CIFAR dataset has two versions: CIFAR 10 and CIFAR 100, where the former has 10 classes, 6,000 images in every class, while the latter has 100 classes with 600 samples in each class. CIFAR 10-LT and CIFAR 100-LT are two long-tailed versions of CIFAR dataset (named semi-synthetic long-tailed datasets in this paper), where the number of samples in each class is determined by a controllable imbalance factor. The commonly used imbalance factors are 10, 50, 100. The test set remains unchanged with even distribution.

**LVIS v0.5** (Gupta et al., 2019) is a large vocabulary instance segmentation dataset with 1231 classes. It contains a 693,958 train set, and a relatively balanced test/ valid set.

**Cora-Full** (Bojchevski & Günnemann, 2018) is a citation network dataset. Each node represents a paper with a sparse bag-of-words vector as the node attribute. The edge represents the citation relationships between two corresponding papers, and the node category represents the research topic.

**Email** (Yin et al., 2017) is a network constructed from email exchanges in a research institution, where each node represents a member, and each edge represents the email communication between institution members.

**Wiki** (Mernyei & Cangea, 2020) is a network dataset of Wikipedia pages, with each node representing a page and each edge denoting the hyperlink between pages.

**Amazon-Clothing** (McAuley et al., 2015) is a product network that contains products in "Clothing, Shoes and Jewelry" on Amazon, where each node represents a product and is labeled with low-level product categories. The node attributes are constructed based on the product's description, and the edges are established based on their substitutable relationship ("also viewed").

**Amazon-Electronics** (McAuley et al., 2015) is another product network constructed from products in "Electronics". The edges are created with the complementary relationship ("bought together") between products.

## C  DETAILS ON EXPERIMENT SETTING

### C.1  HYPERPARAMETER SETTINGS

Here we provide the details of parameter settings. For a fair comparison, we use the default hyperparameter settings for all methods on all datasets. For the ImageNet dataset, Decoupling method uses ResNeXt-50 for 90 epochs training as the backbone, Stochastic Gradient Descent (SGD) optimizer with momentum 0.9, the batch size is 512, and learnable weight scaling (LWS) retrain 10epochs; TDE method uses ResNeXt-50 for 90 epochs training as backbone, the hyperparameter of SGD is set to 0.9, the batch size is 512; MiSLAS method uses ResNet50 and cosine learning rate schedule. For iNaturalist dataset, Decoupling method uses ResNet-152 for 200 epochs training as backbone, SGD optimizer with momentum 0.9, the batch size is 512, LWS retrain 10epochs; TDE method uses ResNeXt-50 for 90 epochs training, SGD optimizer with momentum 0.9, the batch size is 512; MiSLAS method uses ResNet50 and cosine learning rate schedule. For Place365 dataset, Decoupling method uses ResNet-152 for 200 epochs as backbone, SGD optimizer with momentum 0.9, the batch size is set to 512, learnable weight scaling retrain 10epochs; TDE method utilizes ResNeXt-50 for 90epochs, SGD optimizer with momentum 0.9, the batch size is 512; MiSLAS method uses ResNet-152 and cosine learning rate schedule. For Cifar 10-LT and Cifar 100-LT datasets, Decoupling uses ResNet-32 as backbone for 200 epochs training; TDE method uses ResNet-32 for 200 epochs; MiSLAS uses ResNet-32 for 200epoch. For GraphSMOTE, we set the weight of edge reconstruction loss to $1e-6$ as in the original paper. For LTE4G, we adopt the best hyperparameter settings reported in the paper. For Tail-GNN, we set the degree threshold to 5 (*i.e.*, nodes having a degree of no more than 5 are regarded as tail nodes), which is the default value in.

### C.2  EVALUATION METRICS

Considering the long-tailed distribution, accuracy, precision, recall, balanced accuracy, and mean average precision are used as the evaluation metrics.

**Accuracy** (Acc) (Sorower, 2010) provides an overall measure of the model's correctness in predicting the entire test set. Acc favours the majority class as each instance has the same weight and contributes equally to the value of accuracy. In image classification and instance segmentation tasks, besides the overall accuracy across all classes, we follow previous work to comprehensively assess the performance of each method. We calculate the accuracy of three distinct subsets: many-shot classes (classes with over 100 training samples), medium-shot classes (classes with 20 to 100 training samples), and few-shot classes (classes with under 20 training samples). In the task of multi-label learning, each instance can be assigned to multiple classes, making predictions fully correct, partially correct, or fully incorrect. To capture the quality of these predictions, we utilize ACC@$k(k=1,3,5)$ as evaluation metrics, which measure the top-$k$ accuracy.

Table 9: Six metrics for evaluating long-tailed algorithms, where $TP, TN, FP, FN$ stand for true positive, true negative, false positive, false negative, and $AP_i$ is average precision for class $i$. Here we give computations of two-class classification, which are slightly different for different tasks in our benchmark.

| Metric name | Computation | Description |
|---|---|---|
| Acc | $\frac{TP+FN}{TP+TN+FP+FN}$ | Correct predictions of the algorithm on the dataset. |
| Precision | $\frac{TP}{TP+FP}$ | Fraction of correctly predicted positive instances against total positively predicted instances. |
| Recall | $\frac{TP}{TP+FN}$ | Fraction of correctly predicted positive instances against total positively classified instances. |
| bAcc | $\frac{TP/(TP+FN)+TN/(TN+FP)}{2}$ | Arithmetic mean of the recalls for all classes. |
| MAP | $\frac{1}{T}\sum_{i=1}^{T} AP_i$ | Average over $AP$s for all classes. |
| Time | - | Time (training time/inference time) of the algorithm. |

**Precision** (Grandini et al., 2020) measures the number of correctly predicted positive instances against all the instances that were predicted as positive. In this paper, we use macro-precision, which is calculated by averaging the precision scores for each predicted class. The Macro approach considers all the classes equally, regardless of their size, ensuring the effect of majority classes is considered as important as that of minority classes.

**Recall** (Grandini et al., 2020) is a metric that measures the proportion of true positive instances out of all the positive instances. In our evaluation, we utilize macro-recall, which is calculated by averaging recall for each actual class. Notably, recall equals accuracy when the test set is balanced.

**Balanced accuracy** (bAcc) (Grandini et al., 2020) is the arithmetic mean of recall values calculated for each individual class. It is insensitive to imbalanced class distribution, as it treats every class with equal weight and importance and ensures minority classes have a more than proportional influence.

**Mean average precision** (MAP) (Rezatofighi et al., 2019) is a ranking-based metric that is the average of Average Precision (AP) over different classes, where AP is the area under the precision-recall curve.

