# OpenReview forum: "HeroLT: Benchmarking Heterogeneous Long-Tailed Learning"
_ICLR.cc/2024/Conference — ICLR 2024 Conference Withdrawn Submission_

### Official Review · Reviewer_Tkrb · 2023-10-18

**Soundness:** 2 fair
**Presentation:** 3 good
**Contribution:** 2 fair
**Rating:** 3
**Confidence:** 4

**Summary:**

The paper develops long-tailed learning benchmark named HEROLT, which integrates 15 state-of-the-art algorithms and 6 evaluation metrics on 16 real-world benchmark datasets across 5 tasks from 3 domains. They provide the systematic view from three pivotal angles and do extensive experiments.

**Strengths:**

1. The paper is exceptionally well-written, with each part being introduced in a clear and concise manner, making it easy to understand.
2. The paper demonstrates substantial effort, with convincing experimental results.
3. The paper presents a refreshing classification approach, introduces new metrics, and provides reliable benchmarks.

**Weaknesses:**

1. Although a new metric is designed and experimental analysis is conducted, the originality of this work is minimal. It mostly summarizes existing methods with few new designs or practices. It would be more valuable if there were innovative ideas and contributions.
2. The selection of datasets seems extensive, but it is essentially a combination of different domains, without a clear purpose. The data is heterogeneous, while the methods are specialized for different problems. After the integration, specific domain methods are still used for specific domain testing, without transferability or comparison. What is the purpose of this integration?
3. At least in the field of image recognition, these methods cannot be considered as "state-of-the-art methods" as mentioned in the paper. They serve as baselines in recent papers. For example, [1] and [2] are methods with better performance.
4. The finding that "none of the algorithms statistically outperforms others across all tasks and domains, emphasizing the importance of algorithm selection in terms of scenarios" is not surprising but rather intuitive. The subsequent analysis is based on specific dataset experiments, and it remains a mystery how to better guide the design process.

[1] Long-Tailed Recognition via Weight Balancing. CVPR22

[2] Distribution alignment: A unified framework for long-tail visual recognition. CVPR21

**Questions:**

1. What are the specific purposes and implications of Assumption 1 and Assumption 2 in the subsequent context? The use of insufficiently  proof by contradiction is questionable and unconvincing. For example, in the examples of Assumption 1, the feature distribution can be linearly separable. It is confusing to make assumptions about the data without considering the specific training process of the model. If the data possessed such favorable properties, the long-tail distribution problem would not be so challenging. Personally, I believe the difficulties lie in the model's inability to effectively distinguish and generalize features.

2. See others in weakness.

---

> ### Author Response · Authors · 2023-11-20
> **Part 1/2**
>
> >**Q1:** It would be more valuable if there were innovative ideas and contributions.
>
> **A1:** Thank you for your suggestion.
>
> As a benchmark paper, we would like to emphasize that one of our main contributions is the development of a **comprehensive toolbox for heterogeneous long-tailed learning:**
>
> - It facilitates benchmarking heterogeneous long-tailed learning, where the data are of different **types** and thus may have different **pre-processing**.
>
> - Our toolbox integrates the datasets and enables **uniform evaluation** using a standardized function.
>
> - We actively engaged in adapting the **SOTA** long-tailed learning algorithms for multiple tasks (e.g., image classification, instance segmentation).
>
> In addition, we give a **guideline** for dataset and model selection by long-tailed metrics:
>
> - We analyze the potential performance of extreme classification methods, methods for imbalance, long-tail learning, or ordinary machine learning methods when the datasets exhibiting varying degrees of long-tail characteristics.
>
> - Our experimental analysis shows that this guideline can partially explain the experimental results.
>
> &nbsp;
> >**Q2:** The data is heterogeneous, while the methods are specialized for different problems. What is the purpose of this integration?
>
> **A2:** Thank you for pointing this out. We would like to emphasize the following points:
>
> - **Figure 1 illustrates that there is a gap in leveraging multi-model data for building a long-tailed learning algorithm.** However, limited work pays attention to heterogeneous long-tailed learning. Our work establishes a foundational platform for the exploration of this novel area, giving a systematic view of the common challenges, and encouraging the solutions for diverse real-world tasks.
>
> - Our study incorporated specific methods to address the challenges of heterogeneous long-tailed learning. For instance, **datasets like Cora-Full and Email encompass tabular feature information and relational structural information.** Methods such as GraphSMOTE, ImGAGN, TailGNN, and LTE4G effectively exploit both tabular and relational information, and we provide a comparison of these methods. Furthermore, **datasets such as AMAZONCat-13K, Amazon-Clothing, and Amazon-Electronics, sourced from Amazon, span multiple data modalities (sequential, relational, and tabular) while sharing product IDs.** Consequently, concating representations learned from listed algorithms with identical IDs may be a solution to heterogeneous long-tailed learning. However, there exists a research gap in multi-model long-tailed learning algorithms. We propose the first heterogeneous long-tailed benchmark, paving the path for future investigations in this area.
>
> - **We propose a toolbox to facilitate heterogeneous long-tailed learning**, where the data are of different types and thus may have different pre-processing. This integration facilitates standardized evaluation through a uniform evaluation function to furnish the research community with a resource for comparing methods in heterogeneous long-tailed learning.
>
> &nbsp;
> >**Q3:** At least in the field of image recognition, these methods cannot be considered as "state-of-the-art methods" as mentioned in the paper.
>
> >**A3:** Thanks for your valuable suggestion!
>
> - For **multi-label text classification and node classification tasks**, we used state-of-the-art models such as XR-Transformer (2021), XR-Linear (2022), ImGAGN (2021), TailGNN (2021), LTE4G (2022).
>
> - We have added the discussion of the two papers for **image classification and instance segmentation tasks** in Appendix B.2, and are conducting experiments to compare some state-of-the-art methods. The experimental results we have obtained so far are as follows, and we will provide the complete experimental results in our new version.
>
> | Dataset  | Method | Acc|  | | Precision  |  |  | Recal |  |  |MAP  |  |  | Time (s) |
> |-------|--------|:--------:|--------|--------|:--------:|----|---------|:---------:|-------|----|:-------:|--------|--------|:--------:|
> |  |  | 100 | 50 | 10 | 100 |  50 |  10 |  100 |  50 |  10 |  100 |  50 |  10 |  |
> | CIFAR-10-LT  | PaCo   | 85.9 | 88.3 | 91.7 | 86.0 | 88.3 | 91.7 | 85.9 | 88.3 | 91.7 | 75.5 | 79.4 | 85.1 | 4.6 |
> | CIFAR-100-LT | PaCo  | 51.2 | 55.4 | 66.0 | 51.2 | 55.9 | 66.2 | 51.2 | 55.4 | 66.0 | 34.9 | 34.2 | 46.0 | 2.5 |
>
> | Dataset    | Method | Acc |  |   |  | Precision | Recall | MAP  | Time (s) |
> |------------|--------|:------:|--------|------|---------|:----------:|:-------:|:------:|:-------:|
> |  |  | Many | Medium | Few  | Overall |  | (bAcc) |  |  |
> | ImageNetLT | PaCo   | 67.8 | 56.5   | 37.8 | 58.3    | 58.3      | 58.3   | 37.5 | 58.9    |

---

> ### Author Response · Authors · 2023-11-20
> **Part 2/2**
>
> >**Q4:** The subsequent analysis is based on specific dataset experiments, and it remains a mystery how to better guide the design process.
>
> **A4:** Thank you for your valuable feedback. We would like to emphasize the following points:
>
> - In Section 3.4, we conducted an analysis of the building components of each algorithm, such as mixup, meta-learning, and decoupled training. The decoupled learning-based methods OLTR and BALMS exhibited promising experimental results, suggesting their potential efficacy in the context of long-tailed learning.
>
> - However, no single technique consistently outperforms others across all tasks and datasets. **This is the reason we refrain from structuring our taxonomy based on specific components or modules used for benchmarking, like in prior literature.**
>
> - Our benchmark consists of three novel angles (the characterization of data long-tailedness, the data complexity of various domains, and the heterogeneity of emerging tasks) to enrich the understanding of heterogeneous long-tailed data, thereby inspiring future algorithmic research to **fill the gap of multi-model, multi-task long-tail learning algorithms.**
>
> &nbsp;
> >**Q5:** What are the specific purposes and implications of Assumption 1 and Assumption 2 in the subsequent context?
>
> **A5:** Sorry if our prior assumptions cause any misunderstandings. We have updated the content of the assumptions. In general, we made these two assumptions to **satisfy the distinguishability of head categories, tail categories, and noise** `[1]`. Therefore, we illustrate the outcomes of obeying/violating these two assumptions **on the embedding space after great feature engineering and training (https://anonymous.4open.science/r/HeroLT-9746/figs/assumption.pdf).**
>
> |    | $A2$. Compactness Assumption | $\neg{A2}.$ Violation of Compactness Assumption |
> |---|---|---|
> | **$A1$. Smoothness Assumption** | The long-tailed problem is distinguishable, as in Fig. (b) |  Cannot determine whether it is noise data or data from tail class, as in Fig. (d) |
> | **$\neg{A1}$. Violation of Smoothness Assumption** | Cannot depict the decision region of the head, as in Fig. (c) | Cannot distinguish head, tail, and noise, as in Fig. (e) |
>
> `[1]` He, Jingrui. Rare category analysis. Carnegie Mellon University, 2010.

---

> > ### Comment · Reviewer_Tkrb · 2023-11-22
> >
> > Thank you for your response, and I agree that this paper has made a contribution to the community.
> >
> > However, my main concern remains that the motivation is not convincing. The rigid merging of multiple fields without further exploration has limited significance. Additionally, the subsequent experimental conclusions are relatively limited as well. These are not deficiencies that a simple benchmark design can compensate for.
> >
> > I believe that other reviewers also hold similar views to some extent. Therefore, in my opinion, this paper requires further refinement and improvement. I will keep my current score.

---

### Official Review · Reviewer_wsvH · 2023-10-27

**Soundness:** 3 good
**Presentation:** 3 good
**Contribution:** 2 fair
**Rating:** 5
**Confidence:** 5

**Summary:**

This paper provides a systematic view of long-tailed learning with regard to three pivotal angles: (A1) the characterization of data long-tailedness, (A2) the data complexity of various domains, and (A3) the heterogeneity of emerging tasks. It integrates 15 state-of-the-art algorithms and 6 evaluation metrics on 16 real-world benchmark datasets across 5 tasks from 3 domains and proposes an open-source toolbox HEROLT for the long-tailed tasks.

**Strengths:**

1. The paper involves a significant amount of work, and its structure is relatively clear.
2. This paper introduces Pareto-LT Ratio as a new metric for better measuring datasets and gives guidelines for the use of this metric.
3.  It provides a fair and accessible performance evaluation of 15 state-of-the-art methods on multiple benchmark datasets that are contributing to the community

**Weaknesses:**

1. Despite the considerable effort invested in this work, I think that, apart from introducing the Pareto-LT Ratio, the paper falls short of provoking deeper contemplation on the long-tail problem for the readers. Instead, it primarily serves as a means of testing existing methods as baselines across various tasks.
2. I have reservations about the two assumptions mentioned in Section 2.1. Modern deep models' feature space distributions depend on the outcomes of the model learning process, and as a result, these two assumptions may not hold in some cases.
3. I find that the categorization of long-tail tasks in Angle 3 may not be entirely reasonable. For instance, should classification include both multi-label text classification and image classification? And is it appropriate to group single-image and video classification as image classification? Additionally, some other tasks like long-tail semantic segmentation have not been included. Furthermore, certain methods listed in Table 3 may be applicable to multiple tasks; for example, the concept of decoupling could be applied to both classification and detection (segmentation).
4. The citations of methods in the experiments are needed. Meanwhile, some recent methods such as Paco and FCC in the image classification have not been discussed.

[1] Cui, Jiequan, et al. "Parametric contrastive learning." Proceedings of the IEEE/CVF international conference on computer vision. 2021.
[2] Li, Jian, et al. "FCC: Feature Clusters Compression for Long-Tailed Visual Recognition." Proceedings of the IEEE/CVF Conference on Computer Vision and Pattern Recognition. 2023.

**Questions:**

1. What's the meaning of "support region" in the Assumption 1 in Section 2.1?

---

> ### Author Response · Authors · 2023-11-20
> **Part 1/2**
>
> >**Q1:** It primarily serves as a means of testing existing methods as baselines across various tasks.
>
> **A1:** We would like to emphasize that one of our main contributions is the development of **a comprehensive toolbox for heterogeneous long-tailed learning:**
>
> - It facilitates benchmarking heterogeneous long-tailed learning, where the data are of different **types** and thus may have different **pre-processing**.
> - Our toolbox integrates the datasets and enables **uniform evaluation** using a standardized function.
> - We actively engaged in adapting the **SOTA** long-tailed learning algorithms for multiple tasks (e.g., image classification, instance segmentation).
>
> In addition, we give a **guideline** for dataset and model selection by long-tailed metrics:
>
> - We analyze the potential performance of extreme classification methods, methods for imbalance, long-tail learning, or ordinary machine learning methods when the datasets exhibiting varying degrees of long-tail characteristics.
> - Our experimental analysis shows that this guideline can partially explain the experimental results.
>
> &nbsp;
> >**Q2:** Modern deep models' feature space distributions depend on the outcomes of the model learning process, and as a result, these two assumptions may not hold in some cases.
>
> **A2:** Sorry if our prior assumptions cause any misunderstandings. We have updated the content of the assumptions. In general, we made these two assumptions to **satisfy the distinguishability of head categories, tail categories, and noise** `[1]`. Therefore, we illustrate the outcomes of obeying/violating these two assumptions **on the embedding space after great feature engineering and training (https://anonymous.4open.science/r/HeroLT-9746/figs/assumption.pdf).**
>
> |    | $A2$. Compactness Assumption | $\neg{A2}.$ Violation of Compactness Assumption |
> |---|---|---|
> | **$A1$. Smoothness Assumption** | The long-tailed problem is distinguishable, as in Fig. (b) |  Cannot determine whether it is noise data or data from tail class, as in Fig. (d) |
> | **$\neg{A1}$. Violation of Smoothness Assumption** | Cannot depict the decision region of the head, as in Fig. (c) | Cannot distinguish head, tail, and noise, as in Fig. (e) |
>
> `[1]` He, Jingrui. Rare category analysis. Carnegie Mellon University, 2010.
>
> &nbsp;
> >**Q3:** I find that the categorization of long-tail tasks in Angle 3 may not be entirely reasonable.
>
> **A3:** Thank you for pointing this out.
>
> - In our original paper, the 'Classification' task referred specifically to the task of categorizing tabular data, which we have replaced with **'Tabular Classification'** to avoid confusion.
>
> - In Angle 3, we primarily enumerated five common tasks. However, there exist additional long-tailed tasks, such as sentence-level few-shot relation classification, video classification, entity alignment, and others. **We positioned the discussion of these tasks after their most similar common task**, out of concern for the length and brevity of the paper. In addition, we have added references and introductions to semantic segmentation tasks.
>
> - Yes, some methods can be applied to multiple tasks. For example, we give experimental results for BALMS on the image classification and instance segmentation tasks. While the decoupling approach exhibits promise in addressing segmentation challenges, we do not present such results due to the unavailability of the corresponding model or implementation details in the official code.

---

> ### Author Response · Authors · 2023-11-20
> **Part 2/2**
>
> >**Q4:** Some recent methods such as Paco and FCC in the image classification have not been discussed.
>
> **A4:** Thanks for your valuable suggestion! We have added the discussion of **FCC, PaCo and its following work GPaCo** `[1]` for image classification and instance segmentation tasks in Appendix B.2. In addition, we have added several baselines for image classification and instance segmentation tasks. The experimental results we have obtained so far are as follows, and we will provide the complete experimental results in our new version.
>
> | Dataset  | Method | Acc|  | | Precision  |  |  | Recal |  |  |MAP  |  |  | Time (s) |
> |-------|--------|:--------:|--------|--------|:--------:|----|---------|:---------:|-------|----|:-------:|--------|--------|:--------:|
> |  |  | 100 | 50 | 10 | 100 |  50 |  10 |  100 |  50 |  10 |  100 |  50 |  10 |  |
> | CIFAR-10-LT  | PaCo   | 85.9 | 88.3 | 91.7 | 86.0 | 88.3 | 91.7 | 85.9 | 88.3 | 91.7 | 75.5 | 79.4 | 85.1 | 4.6 |
> | CIFAR-100-LT | PaCo  | 51.2 | 55.4 | 66.0 | 51.2 | 55.9 | 66.2 | 51.2 | 55.4 | 66.0 | 34.9 | 34.2 | 46.0 | 2.5 |
>
> | Dataset    | Method | Acc |  |   |  | Precision | Recall | MAP  | Time (s) |
> |------------|--------|:------:|--------|------|---------|:----------:|:-------:|:------:|:-------:|
> |  |  | Many | Medium | Few  | Overall |  | (bAcc) |  |  |
> | ImageNetLT | PaCo   | 67.8 | 56.5   | 37.8 | 58.3    | 58.3      | 58.3   | 37.5 | 58.9    |
>
> `[1]` Cui, Jiequan, et al. "Generalized parametric contrastive learning." IEEE Transactions on Pattern Analysis and Machine Intelligence (2023).
>
> &nbsp;
> >**Q5:** What's the meaning of "support region" in the Assumption 1 in Section 2.1?
>
> **A5:** Sorry for the confusion. We would like to refer to the decision region `[1]`, a conventional notion in classification problems, where the input data points correspond to a unique output class. For categories $c$, the decision region is defined as
> $S_c=\{x_i : f(x_i)=c, i=1,\ldots,n \}$
>
> `[1]` Gibson, Gavin J., and Colin FN Cowan. "On the decision regions of multilayer perceptrons." Proceedings of the IEEE, 1990.

---

### Official Review · Reviewer_BsKP · 2023-10-31

**Soundness:** 2 fair
**Presentation:** 2 fair
**Contribution:** 2 fair
**Rating:** 3
**Confidence:** 4

**Summary:**

The paper builds a comprehensive benchmark for the long-tail problem, containing multiple datasets and algorithms.

**Strengths:**

1. The long tail problem is a practical and valuable research topic.
2. Benchmark contains different data types and tasks.

**Weaknesses:**

1. Most of the datasets and tasks seem to have been well constructed and the paper seems to merely concatenate them together without much additional integration.
2. The methods discussed in the paper appear to be somewhat dated. For instance, the image classification data that is prominently featured in Table 2 is based on methodologies that were largely developed before 2020. Considering that it is already near the end of 2023, this renders the insights provided by the paper less credible.
3. Referring to section C.1 of the paper, there is a misalignment in the model architectures and the number of epochs used when comparing different methods, which calls into question the article's claim of a 'fair evaluation'.
4. According to the four-point analysis in the upper half of the third page, the Pareto-LT Ratio still requires integration with the Gini coefficient and IF to describe the dataset. The former primarily represents the number of categories, while the latter two indicate data imbalance. So why not directly use the number of categories as a descriptor? Additionally, I do not find comprehensive experimental and theoretical evidences to support this four-point analysis.

**Questions:**

See above.

---

> ### Author Response · Authors · 2023-11-20
> **Part 1/1**
>
> > **Q1:** Most of the datasets and tasks seem to have been well constructed and the paper seems to merely concatenate them together without much additional integration.
>
> **A1:** We would like to emphasize that one of our main contributions is the development of a **comprehensive toolbox for heterogeneous long-tailed learning**:
>
> - It facilitates benchmarking heterogeneous long-tailed learning, where the data are of different **types** and thus may have different **pre-processing**.
>
> - Our toolbox integrates the datasets and enables **uniform evaluation** using a standardized function.
>
> - We actively engaged in adapting the **SOTA** long-tailed learning algorithms for multiple tasks (e.g., image classification, instance segmentation).
>
> - In addition, we give a **guideline** for dataset and model selection by long-tailed metrics by analyzing the potential performance of extreme classification methods, methods for imbalance, long-tail learning, or ordinary machine learning methods when the datasets exhibiting varying degrees of long-tail characteristics.
>
> &nbsp;
> >**Q2:** The methods discussed in the paper appear to be somewhat dated.
>
> **A2:** Thanks for your valuable suggestion!
>
> - For **multi-label text classification and node classification tasks**, we used state-of-the-art models such as XR-Transformer (2021), XR-Linear (2022), ImGAGN (2021), TailGNN (2021), LTE4G (2022).
>
> - We have added the discussion of several recent papers `[1-4]` for **image classification and instance segmentation tasks** in Appendix B.2, and are conducting experiments to compare some of these state-of-the-art methods. The experimental results we have obtained so far are as follows, and we will provide the complete experimental results in our new version.
>
> | Dataset  | Method | Acc|  | | Precision  |  |  | Recal |  |  |MAP  |  |  | Time (s) |
> |-------|--------|:--------:|--------|--------|:--------:|----|---------|:---------:|-------|----|:-------:|--------|--------|:--------:|
> |  |  | 100 | 50 | 10 | 100 |  50 |  10 |  100 |  50 |  10 |  100 |  50 |  10 |  |
> | CIFAR-10-LT  | PaCo   | 85.9 | 88.3 | 91.7 | 86.0 | 88.3 | 91.7 | 85.9 | 88.3 | 91.7 | 75.5 | 79.4 | 85.1 | 4.6 |
> | CIFAR-100-LT | PaCo  | 51.2 | 55.4 | 66.0 | 51.2 | 55.9 | 66.2 | 51.2 | 55.4 | 66.0 | 34.9 | 34.2 | 46.0 | 2.5 |
>
> | Dataset    | Method | Acc |  |   |  | Precision | Recall | MAP  | Time (s) |
> |------------|--------|:------:|--------|------|---------|:----------:|:-------:|:------:|:-------:|
> |  |  | Many | Medium | Few  | Overall |  | (bAcc) |  |  |
> | ImageNetLT | PaCo   | 67.8 | 56.5   | 37.8 | 58.3    | 58.3      | 58.3   | 37.5 | 58.9    |
>
> `[1]` Li, Jian, et al. "FCC: Feature Clusters Compression for Long-Tailed Visual Recognition." CVPR, 2023.
>
> `[2]` Cui, Jiequan, et al. "Generalized parametric contrastive learning." IEEE PAMI, 2023.
>
> `[3]` Alshammari, Shaden, et al. "Long-tailed recognition via weight balancing." CVPR, 2022.
>
> `[4]` Zhang, Songyang, et al. "Distribution alignment: A unified framework for long-tail visual recognition." CVPR 2021.
>
> &nbsp;
> >**Q3:** There is a misalignment in the model architectures and the number of epochs used when comparing different methods.
>
> **A3:** Thank you for pointing this out. In our experiments, we used the default model architecture and hyperparameters for all methods.
>
> - Various methods do employ divergent model structures (e.g., ResNet-32, ResNeXt-50, ResNet-152). However, changes to the model architecture may disrupt the efficacy of the original optimized hyperparameters.
>
> - Furthermore, the difference in model complexity necessitated a distinct number of epochs. This is because using the same number of epochs may result in scenarios where some models reach convergence while others remain inadequately trained. Thus, our benchmark follows the settings of the original papers.
>
> &nbsp;
> >**Q4:** The Pareto-LT Ratio still requires integration with the Gini coefficient and IF to describe the dataset.
>
> **A4:** Thank you for your comment!
> - **Pareto-LT Ratio itself serves as a metric tailored to measure long-tailedness.** Intuitively, the higher the skewness of the data distribution, the higher Pareto-LT Ratio may be; the more number of categories, the higher the value of Pareto-LT Ratio. On the other hand, IF and Gini coefficient are designed to measure imbalance, with the number of categories reflecting solely the size of categories.
>
> - Sample size, the number of categories, IF, Gini coefficient, and Pareto-LT Ratio can provide insights into model design. **Our study concurrently analyzes multiple metrics to enable a more comprehensive characterization of datasets and to inspire potential model design.** However, no metric can comprehensively and accurately depict all the characteristics of a dataset currently. Our analysis may indeed exhibit limitations in specific datasets.

---

> ### Comment · Reviewer_BsKP · 2023-11-20
>
> My concerns remain.
>
> 1. The datasets utilized are well-established and commonly employed. Despite being amalgamated in the paper, there remains a disjointed nature between various tasks.
>
> 2. As a benchmark paper, the lack of alignment in model architectures raises doubts about the reference value of the results.
>
> 3. Upon receiving the email, I immediately checked the author's response and the updates to the paper. However, I could not find any discussion or comparison in the paper regarding the works the authors claimed to “have added” in their response.

---

> > ### Author Response · Authors · 2023-11-21
> >
> > We are conducting experiments on the new-added methods, and the comparison results will be updated in our paper when completed. The current version of the paper mainly contains changes to 2.1 Preliminaries and Problem Definition and B.2 HeroLT Algorithm List.
> >
> > In addition, we have updated the results of PaCo on ImageNet-LT using the aligned backbone. In our future investigations, we will search for the optimal hyperparameters for the few methods that do not use the same backbone and report the results.

---

### Official Review · Reviewer_aihc · 2023-10-31

**Soundness:** 2 fair
**Presentation:** 2 fair
**Contribution:** 2 fair
**Rating:** 5
**Confidence:** 4

**Summary:**

This paper systematically investigates the long-tailed problem in various scenarios. It proposes a heterogeneous view from the aspect of 1) long-tailedness or extreme categories; 2) data complexity of various domains; and 3) heterogeneous tasks. Different from the previous metric (Imbalanced factor, gini coefficient), it proposes a new metric (Pareto-LT) to help measure the extreme number of categories. By conducting empirical studies on various long-tailed benchmarks, the authors provide some insights that different methods may work for different tasks with different long-tailed properties.

**Strengths:**

This paper is well-written, and I think it will contribute to the long-tailed learning community.

1. As far as I know, this is the first work that systematically benchmarks long-tailed learning problems. Moreover, it categorizes long-tailed learning problems from different angles, including the data imbalance/extremes, data types/domains, and applied tasks. It almost covers recent important long-tailed learning challenges.
2. This paper proposes a novel metric, the Pareto-LT Ratio, for measuring both the degree of imbalance and the extreme number of categories of long-tailed datasets.
3. This paper conducts empirical studies on multiple datasets and compares popular methods. It provides valuable insights regarding which types of methods may work on a certain type of task.
4. The authors open-source a toolbox for evaluating long-tailed datasets.

**Weaknesses:**

1. The problem definition in Sec. 2.1 is under an ideal situation. The authors present two assumptions, i.e., the Smoothness Assumption and the Compactness Assumption, which indicate that all data points are annotated and clustered well. However, in real-world scenarios, this might not be easily achieved. A category may contain multiple subcategories, which may obey Assumption 1. Also, the data might contain feature or label noise, making it hard to extract distinctive representations, thereby obeying Assumption 2. The authors should either consider these real-world problems, otherwise declare that they just define long-tailed learning under an ideal situation.
2. The definition of long-tailed learning (Problem 1) seems trivial. It is similar to the definition of supervised learning, except that you emphasize the importance of both head and tail categories. It is better to illustrate the long-tailed property, i.e., $\mathbb{P}(\mathcal{Y})$ obeys a long-tailed distribution, or $\mathcal{Y}$ has an extreme cardinality.
3. The imbalance factor (IF) and Gini coefficient seem to perform a similar effect. The authors mainly consider them equally when compared with the Pareto-LT Ratio. Since the IF metric is more widely used, it seems that the Gini coefficient is less important. I suggest the authors discuss more about the difference between IF and Gini coefficient.

**Questions:**

1. In Fig. 2(a), what is the meaning of the Pareto curve?
2. Why do you choose 20% as a threshold for the Pareto-LT Ratio metric?
3. I find the performance of OLTR on CIFAR-10-LT and CIFAR-100-LT too well. It almost surpasses all long-tailed learning methods using ResNet-32 (https://paperswithcode.com/task/long-tail-learning). However, I failed to find the reproduced results of OLTR on CIFAR-LT in previous works. I wonder if there are some mistakes in this paper.

---

> ### Author Response · Authors · 2023-11-19
> **Part 1/2**
>
> >**Q1:** The authors present two assumptions, i.e., the Smoothness Assumption and the Compactness Assumption. However, in real-world scenarios, this might not be easily achieved.
>
> **A1:** Thank you for your constructive comments.
>
> **Firstly,** we would like to confirm if you are referring to 'A category may contain multiple subcategories, which may *violate* Assumption 1.' If this is the case, we would like to clarify that **our paper does not consider hierarchical categories.** Note that the majority of the long-tailed learning research does not explicitly consider subcategory categorization nor provide systematic evaluation `[1-7]`. As a benchmark paper integrating state-of-the-art methods across multiple domains, we follow the common problem setting.
>
> **Secondly,** **Assumption 2 does not assume a zero-noise scenario.** In real scenarios, data exhibits feature noise and label noise, thus we give Assumption 2 to ensure that the tail categories are distinguishable from the rest of the categories and background noise following `[8]`.
>
> **Finally,** we have updated the content of the assumptions to avoid confusion and will add it to our later version. In general, **we made these two assumptions to satisfy the distinguishability of head categories, tail categories, and noise**. Therefore, we illustrate the outcomes of obeying/violating these two assumptions on the embedding space after great feature engineering and training (**https://anonymous.4open.science/r/HeroLT-9746/figs/assumption.pdf**).
>
> |    | $A2$. Compactness Assumption | $\neg{A2}.$ Violation of Compactness Assumption |
> |---|---|---|
> | **$A1$. Smoothness Assumption** | The long-tailed problem is distinguishable, as in Fig. (b) |  Cannot determine whether it is noise data or data from tail class, as in Fig. (d) |
> | **$\neg{A1}$. Violation of Smoothness Assumption** | Cannot depict the decision region of the head, as in Fig. (c) | Cannot distinguish head, tail, and noise, as in Fig. (e) |
>
> `[1]` Zhang, Jiong, et al. "Fast multi-resolution transformer fine-tuning for extreme multi-label text classification." NeurIPS, 2021.
>
> `[2]` Zhong, Zhisheng, et al. "Improving calibration for long-tailed recognition." CVPR, 2021.
>
> `[3]` Zhou, Boyan, et al. "Bbn: Bilateral-branch network with cumulative learning for long-tailed visual recognition." CVPR, 2020.
>
> `[4]` Kang, Bingyi, et al. "Decoupling representation and classifier for long-tailed recognition." ICLR, 2020.
>
> `[5]` Zhao, Tianxiang, Xiang Zhang, and Suhang Wang. "Graphsmote: Imbalanced node classification on graphs with graph neural networks." WSDM, 2021.
>
> `[6]` Qu, Liang, et al. "Imgagn: Imbalanced network embedding via generative adversarial graph networks." KDD, 2021.
>
> `[7]` Yun, Sukwon, et al. "Lte4g: long-tail experts for graph neural networks." CIKM, 2022.
>
> `[8]` He, Jingrui. Rare category analysis. Carnegie Mellon University, 2010.
>
> &nbsp;
> >**Q2:** The definition of long-tailed learning (Problem 1) seems trivial. It is better to illustrate the long-tailed property.
>
> **A2:** Thanks for your suggestion! We would like to clarify that long-tailed data exhibits a highly skewed data distribution and an extensive number of categories, and we give some examples with long-tailed distributions in Appendix A.
>
> For the sake of clarity, we have modified the Long-Tailed Learning problem as follows.
>
> **Given:** a training set $\mathcal{D}$ of $n$ samples from $C$ distinct categories and the label set $\mathcal{Y}$. **The data follows long-tailed distribution, i.e., the frequency of the categories can be approximated as $\lim_{y\rightarrow \ \infty}\ e^{ty}\ P(Y>y)=\infty$, for all $t>0$, where $Y$ is a random variable.**
>
> **Find:** a function $f: \mathcal{X} \rightarrow \mathcal{Y}$ that gives accurate label predictions on both head and tail categories.
>
> &nbsp;
> >**Q3:** I suggest the authors discuss more about the difference between IF and Gini coefficient.
>
> **A3:** Thanks for pointing this out. We will add more discussion in Appendix B.1:
>
> **Imbalance Factor: $n_1/n_C$.**
>
> **Gini Coefficient: $\sum_{i=1}^{C}\sum_{j=1}^{C}|n_i-n_j| / 2nC$.**
>
> Both IF and Gini coefficient try to quantify data imbalance, while IF measures the size of the most majority to the size of the most minority categories, and Gini coefficient considers the difference between the sizes of any two categories, less affected by extreme samples or absolute data size.
>
> &nbsp;
> >**Q4:** In Fig. 2(a), what is the meaning of the Pareto curve?
>
> **A4:** Thank you for pointing this out! The curve is **the probability density function of the Pareto distribution** which is a skewed distribution with a long tail, i.e., much of the data is in the tails `[1]`.
>
> `[1]` Arnold, Barry C. "Pareto distribution." Wiley StatsRef: Statistics Reference Online (2014): 1-10.

---

> ### Author Response · Authors · 2023-11-19
> **Part 2/2**
>
> > **Q5:** Why do you choose 20% as a threshold for the Pareto-LT Ratio metric?
>
> **A5:** We choose $p$=20% by default according to **Pareto principle (the 80-20 rule)** `[1]`, i.e., the top 20% categories contain 80% of samples.
>
> Meanwhile, two long-tailed datasets are comparable if using the same $p$ value. However, if the user has a unique definition of head categories in a particular domain, a different value of $p$ can be used.
>
> `[1]` Dunford, Rosie, Quanrong Su, and Ekraj Tamang. "The pareto principle." (2014).
>
> &nbsp;
> > **Q6:** I failed to find the reproduced results of OLTR on CIFAR-LT in previous works.
>
> **A5:** Thank you for checking.
> - OLTR was not originally implemented on the CIFAR dataset and was only implemented on the ImageNet and Places datasets. We changed the dataset object defined in the official code, **used the default settings for Places to conduct the two-stage training based on ResNet-152**, and then reported the test results.
>
> - The great performance of OLTR on the CIFAR datasets may be related to the model structure. Our past experimental results show that **the ResNet-152-based OLTR method outperforms the ResNet-10-based OLTR in accuracy.**
>
> |              | IF:        | 100  | 50   | 10   |
> |--------------|-----------|------|------|------|
> | CIFAR-10-LT  | ResNet-10  | 50.7 | 64.2 | 83.0 |
> |              | ResNet-152 | 88.7 | 90.8 | 93.3 |
> | CIFAR-100-LT | ResNet-10  | 56.5 | 63.7 | 77.2 |
> |              | ResNet-152 | 67.6 | 72.0 | 77.2 |

---

> > ### Comment · Reviewer_aihc · 2023-11-20
> >
> > For Q1, what I mean is that a category may contain multiple sub-populations, but annotated as the same category. In the embedding space, their features may be distributed across multiple clusters (as illustrated in Figure 2(b) in the main paper). I think this situation is common in the real scenarios.
> >
> > For Q6, your results are reproduced with ResNet-10 and ResNet-152. However, in Appendix C.1, you use ResNet-32 for Decoupling, TDE, and MiSLAS on CIFAR. Therefore I am concerned with the fairness in the comparison.
> >
> > And your answers to Q2-Q5 have addressed my concerns. Thanks for your response.

---

> ### Author Response · Authors · 2023-11-21
>
> >**Q1:** For Q1, what I mean is that a category may contain multiple sub-populations, but annotated as the same category. In the embedding space, their features may be distributed across multiple clusters (as illustrated in Figure 2(b) in the main paper). I think this situation is common in the real scenarios.
>
> **A1:** Yes, a category may potentially contain multiple sub-populations. But it does not contradict our Assumption 1.
>
> - Firstly, **we would like to clarify the notion of smoothness is defined based on the well-trained embedding space instead of the input feature space.** For the head categories with rich data and labels,  with sufficient training, the examples should be well-clustered together with a smooth decision region in the learned embedding space.
>
> - Secondly, **We consider smoothness to be a continuous quantitative measure**, different degrees of smoothness can impair the performance of long-tail learning to varying degrees, but do not necessarily lead to a complete failure of the algorithm. When the degree of smoothness is low, as shown on the left (https://anonymous.4open.science/r/HeroLT-9746/figs/example.pdf), the decision region of the head category may be difficult to depict smoothly and the tail category may be difficult to classify, especially tail category has fewer samples. When relatively smooth, as shown on the right, the tail category may be easier to recognize.
>
> &nbsp;
> >**Q2:** For Q6, your results are reproduced with ResNet-10 and ResNet-152. However, in Appendix C.1, you use ResNet-32 for Decoupling, TDE, and MiSLAS on CIFAR. Therefore I am concerned with the fairness in the comparison.
>
> **A2:** We understand your concern.
>
> - We tried to adopt the same backbone for all methods in the image classification task. However, given the existence of additional hyperparameters to these methods, only adjusting the model backbone may compromise model performance. Consequently, in our reported outcomes, we used the default settings.
>
> - In our future investigations, we will search for the optimal hyperparameters for the small number of methods that do not use the same backbone and report the results.

---

> > ### Comment · Reviewer_aihc · 2023-11-21
> >
> > Thank you for your feedback.
> >
> > First, your assumptions are based on "well-trained embedding space". Although it is acceptable in the paper, it will compromise its practicality.
> >
> > Second, it is unreasonable to use different backbones on the same dataset. This will lead to unfair comparisons and raise concerns about your conclusions.
> >
> > Therefore, I have to decline my score, considering the unconvincing results. Nevertheless, I believe that your work has a certain contribution, and I encourage the authors to improve your paper in the future.